# Spinal V1 inhibitory interneuron clades differ in birthdate, projections to motoneurons, and heterogeneity

Andrew E Worthy[1,2], Joanna T Anderson[2], Alicia R Lane[2], Laura J Gomez-Perez[2], Anthony A Wang[1], Ronald W Griffith[1,2], Andre F Rivard[2], Jay B Bikoff[3], Francisco J Alvarez[1,2]*

[1]Department of Physiology, Emory University School of Medicine, Atlanta, United States; [2]Department of Cell Biology, Emory University School of Medicine, Atlanta, United States; [3]Department of Developmental Neurobiology, St Jude Children's Research Hospital, Memphis, United States

## eLife Assessment

This study provides a **valuable** description of subtypes of V1 neurons, including birthdates and connections to motor neurons. V1 neurons are one of the main groups of inhibitory neurons in the spinal cord. The methods of data collection and analysis are **convincing**. This work will interest developmental biologists and neuroscientists working on spinal circuits.

**\*For correspondence:**
francisco.j.alvarez@emory.edu

**Competing interest:** The authors declare that no competing interests exist.

**Abstract** Spinal cord interneurons play critical roles shaping motor output, but their precise identity and connectivity remain unclear. Focusing on the V1 interneuron cardinal class we defined four major V1 subsets in the mouse according to neurogenesis, genetic lineage-tracing, synaptic output to motoneurons, and synaptic inputs from muscle afferents. Sequential neurogenesis delineates different V1 subsets: two early born (Renshaw and Pou6f2) and two late born (Foxp2 and Sp8). Early born Renshaw cells and late born Foxp2-V1 interneurons are tightly coupled to motoneurons, while early born Pou6f2-V1 and late born Sp8-V1 interneurons are not, indicating that timing of neurogenesis does not correlate with motoneuron targeting. V1 clades also differ in cell numbers and diversity. Lineage labeling shows that the Foxp2-V1 clade contains over half of all V1 interneurons, provides the largest inhibitory input to motoneuron cell bodies, and includes subgroups that differ in birthdate, location, and proprioceptive input. Notably, one Foxp2-V1 subgroup, defined by postnatal Otp expression, is positioned near the LMC and receives substantial input from proprioceptors, consistent with an involvement in reciprocal inhibitory pathways. Combined tracing of ankle flexor sensory afferents and interneurons monosynaptically connected to ankle extensors confirmed placement of Foxp2-V1 interneurons in reciprocal inhibitory pathways. Our results validate previously proposed V1 clades as unique functional subtypes that differ in circuit placement, with Foxp2-V1 cells forming the most heterogeneous subgroup. We discuss how V1 organizational diversity enables understanding of their roles in motor control, with implications for their diverse ontogenetic and phylogenetic origins.

## Introduction

The spinal cord contains a diversity of interneurons which lend to its vast computational power. Many of these interneurons are inhibitory and critically modulate and pattern motoneuron firing adjusting the timing and force of muscle contractions. We focus here on one major group of ventral inhibitory

interneurons known as V1 that originate from p1 progenitors, express the transcription factor (TF) Engrailed-1 (En1), and send ipsilateral axons throughout the ventral horn including lamina IX (LIX), where they densely innervate motoneuron cell bodies and proximal dendrites (*Alvarez et al., 2005*; *Goulding, 2009*). This early study proposed a diversity of phenotypes and circuit roles for V1 interneurons based on differential expression of calcium buffering proteins, location, and synaptology. Electrophysiological and modeling studies then showed that V1 interneurons play crucial roles shaping motor output and modulating locomotor speed, governing flexion-extension at the level of central pattern generator half-centers and/or last-order reciprocal inhibition of antagonistic motoneurons and providing recurrent feedback inhibition of motoneuron firing (*Sapir et al., 2004*; *Zhang et al., 2014*; *Britz et al., 2015*; *Falgairolle and O'Donovan, 2019*; *Falgairolle and O'Donovan, 2021*; *Shevtsova et al., 2022*). However, the relations between these multiple functions and V1 heterogeneity are not fully clear. Accordingly, there is significant interest in defining the molecular identity, circuit organization, and motoneuron connections of different V1 interneuron subtypes. This is of further significance because recent findings suggest that early disconnection of V1 synapses from motoneurons contributes to dysfunction and presages motoneuron death in amyotrophic lateral sclerosis (ALS) (*Wootz et al., 2013*; *Salamatina et al., 2020*; *Allodi et al., 2021*; *Mora et al., 2024*; *Montañana-Rosell et al., 2024*).

Prior work revealed that V1 interneurons are organized into at least four major clades according to their positions and molecular identity at postnatal day 0 (P0) (*Bikoff et al., 2016*). Each clade was defined by expression of unique TFs. Co-expression of V-maf musculoaponeurotic fibrosarcoma oncogene homologs A and B (MafA/MafB) defines the V1 clade that corresponds with Renshaw cells (*Benito-Gonzalez and Alvarez, 2012*; *Stam et al., 2012*; *Bikoff et al., 2016*). The other three clades are respectively defined by expression at P0 of the TFs P.O.U. domain class 6 homeobox 2 (Pou6f2), Forkhead box P2 (Foxp2), and Specificity protein 8 (Sp8) (*Bikoff et al., 2016*). Within these three clades, additional diversity was uncovered by further combinations of TF expression and positions (*Bikoff et al., 2016*; *Gabitto et al., 2016*; *Sweeney et al., 2018*). A recent harmonized atlas of several mouse spinal cord transcriptomic studies (*Russ et al., 2021*) identified seven possible V1 groups, including Renshaw cells, Pou6f2-V1s, and three Foxp2-V1 groups. V1 genetic diversity parallels the functional diversity described in physiological and modeling studies, but whether V1 clades occupy specific functional niches in spinal motor circuits remains unclear, partly because of a lack of information about their synaptic inputs and outputs and their origins.

Here, we clarify the origins, diversity, and synaptic relations with motoneurons of four major V1 clades. V1 neurogenesis was previously divided into early (E9.5 to E10.5) and late phases (E11 to E12.5), each producing distinct V1 interneurons (*Benito-Gonzalez and Alvarez, 2012*). It is now well accepted that temporal and spatial properties intersect during neurogenesis to create cellular diversity from each spinal cord progenitor domain (*Sagner, 2024*; *Deska-Gauthier et al., 2020*; *Deska-Gauthier and Zhang, 2021*; *Osseward et al., 2021*; *Sagner et al., 2021*; *Deska-Gauthier et al., 2024*). Using nodal intersections between transcriptomics and neurogenesis, one report described seven embryonic V1 groups defined by TFs with temporally restricted expression (*Delile et al., 2019*). However, because of the dynamic nature of TF expression in spinal interneurons, it is difficult to match embryonic TFs to V1 clades previously defined at P0. Using clade-defining TFs in combination with 5-ethynyl-2'-deoxyuridine (EdU) birthdating, we identified a relationship between neurogenesis timing and clade identity and uncovered additional diversity within Foxp2-V1 interneurons, the largest V1 clade. To further study Foxp2-V1 interneurons, we used intersectional genetics to lineage-label their cell bodies and axons. This revealed subdivisions according to location and expression of additional TFs. We also found that Foxp2-V1s establish the highest density of V1 synapses on limb-related lateral motor column (LMC) motoneurons. They also include subgroups receiving dense proprioceptive inputs, with some having connectivity typical of reciprocal IaINs. Together with Renshaw cells, they contribute the majority of V1 synapses on motoneuron cell bodies and proximal dendrites, while synapses from other clades (Pou6f2 and Sp8) have minimal representation. We conclude that V1 clades differ in time of neurogenesis, internal heterogeneity, and synaptic targeting of motoneurons.

## Results

### V1 interneurons belonging to different clades have distinct timing of neurogenesis

To examine the birthdates of the different V1 clades, we lineage traced all V1 interneurons using $En1^{cre/+}$, $Ai9\ R26^{lsl-tdTomato}$ mice (sometimes intersected with $Foxp2^{flpo/+}$, $R26^{RCE:dual-EGFP}$) and pulse-labeled developing embryos by injecting pregnant females with EdU at different 12 hr intervals from E9.5 to E12.5 (*Figure 1A*). Pups were analyzed at P5. This age was chosen to maximize expression of V1 clade defining TFs (MafB, Pou6f2, Foxp2, and Sp8) for antibody detection. We analyzed spinal cords at E9.5 (n=3; 3 litters), E10 (n=6; 3 litters), E10.5 (n=5; 3 litters), E11 (n=5; 2 litters), E11.5 (n=6; 3 litters), E12 (n=3; 1 litter), and E12.5 (n=4; 2 litters). In all animals, we confirmed the expected ventrolateral to dorsomedial sequence of cell birthdates in the mammalian spinal cord (*Figure 1B*; *Altman and Bayer, 2001*). One animal pulse-labeled at E11 was removed from the analyses because the pattern of EdU labeling did not correspond to the expected distribution (marked with x in *Figure 1D*). We defined positive neurons as those with more than two-thirds of the nucleus showing homogeneous EdU fluorescence to ensure only V1 cells in S-phase at the time of injection were included in analyses. Neurons with speckles or partial nuclear labeling (*Figure 1C*) may arise from EdU dilution in successive division cycles, or incorporation of EdU during very late S-phase or DNA repair events (*Packard et al., 1973*; *Ferreira et al., 1997*; *Taupin, 2007*).

Using our stringent criteria for EdU+ cell classification, we found different percentages of V1s generated at different time points. Peak V1 neurogenesis occurred at E11 with 23.4%±2.9 (mean ± SD) of V1s incorporating EdU (*Figure 1D and E*). After adding together all EdU-labeled V1s at all time points, we account for 71.4% of all V1 interneurons. If we include V1 neurons with speckles or partial nuclear labeling, we overrepresent V1s by more than double (255.9%) and obscure differences in neurogenesis timing. Nevertheless, some overlap occurs among animals' pulse-labeled in contiguous 12 hr time points. Given the fast diffusion and elimination of EdU in mice (1.4±0.7 and 24.1±2.9 min half-times, respectively; *Cheraghali et al., 1994*) there should be little overlap in EdU bioavailability with injections separated by 12 hr, but other limitations can reduce the effective time resolution of the technique. In our mating protocol, fertilization can occur anytime within a 12 hr window (see Materials and methods). It is also common within single litters to find animals with developmental differences of 6–12 hr. Finally, S-phase duration can vary between 4 and 17 hr depending on embryonic stage and progenitor type (*Ponti et al., 2013*). All these conditions create opportunities for labeling overlaps after EdU injections spaced by 12 hr intervals. We overcame time-resolution issues by using rigorous criteria for defining EdU labeling and using large sample sizes that included animals from different pregnancies at each time point. This allowed robust birthdate estimates, even within 12 hr intervals. Inter-animal variability was most noted at times around peak neurogenesis (E10.5 and E11.5, *Figure 1E*). This is best explained by rapid acceleration and deceleration of V1 generation at these times.

To analyze birthdates of V1 interneurons identified by clade-specific markers (*Figure 1F–I*), we first compared the number of V1s expressing clade-defining TFs at P5 with previous estimates at P0 (*Bikoff et al., 2016*). Antibody specificities are shown in *Figure 1—figure supplement 1*, in *Figure 1—figure supplement 2*, and in a previous paper (*Bikoff et al., 2016*). Overall, V1s expressing MafB at P5 were 9.4%±1.6 (± SD) of the whole V1 population (n=21 animals) which differs from the 25% estimate at P0 from *Bikoff et al., 2016*. Smaller differences were found for Pou6f2+ V1s (8.1%±4.6 [n=19] compared to 13%) Foxp2+ V1s (32.5%±8.4 [n=26] compared to 34%) and Sp8+ V1s (8.8%±2.6 [n=26] compared to 13%). These small differences are attributable to differences in age and/or immunocytochemical (ICC) sensitivity. In contrast, the large differences found in multi-clade MafB-V1s likely result from rapid downregulation of MafB from P0 to P5 in some V1 clades. MafB is expressed at P0 in three V1 clades (*Bikoff et al., 2016*): ventral MafA-calbindin Renshaw cells, subpopulations of dorsally located Pou6f2-V1s, and subgroups of Foxp2-V1s distributed throughout the ventral horn. MafB is quickly downregulated after birth in Foxp2-V1s, but expression is maintained in Pou6f2-V1s and ventral Renshaw cells (see below). When considering only ventral MafB-V1s located in the Renshaw area (Renshaw V1 clade), we obtained identical percentages at P5 and P0 for the MafA-Renshaw cell V1 clade (5.5%±1.7, n=21 vs 5.0%; *Bikoff et al., 2016*). We conclude that TF ICC detection at P5 provides an accurate sampling of V1 clades previously defined at P0 (*Bikoff et al., 2016*).

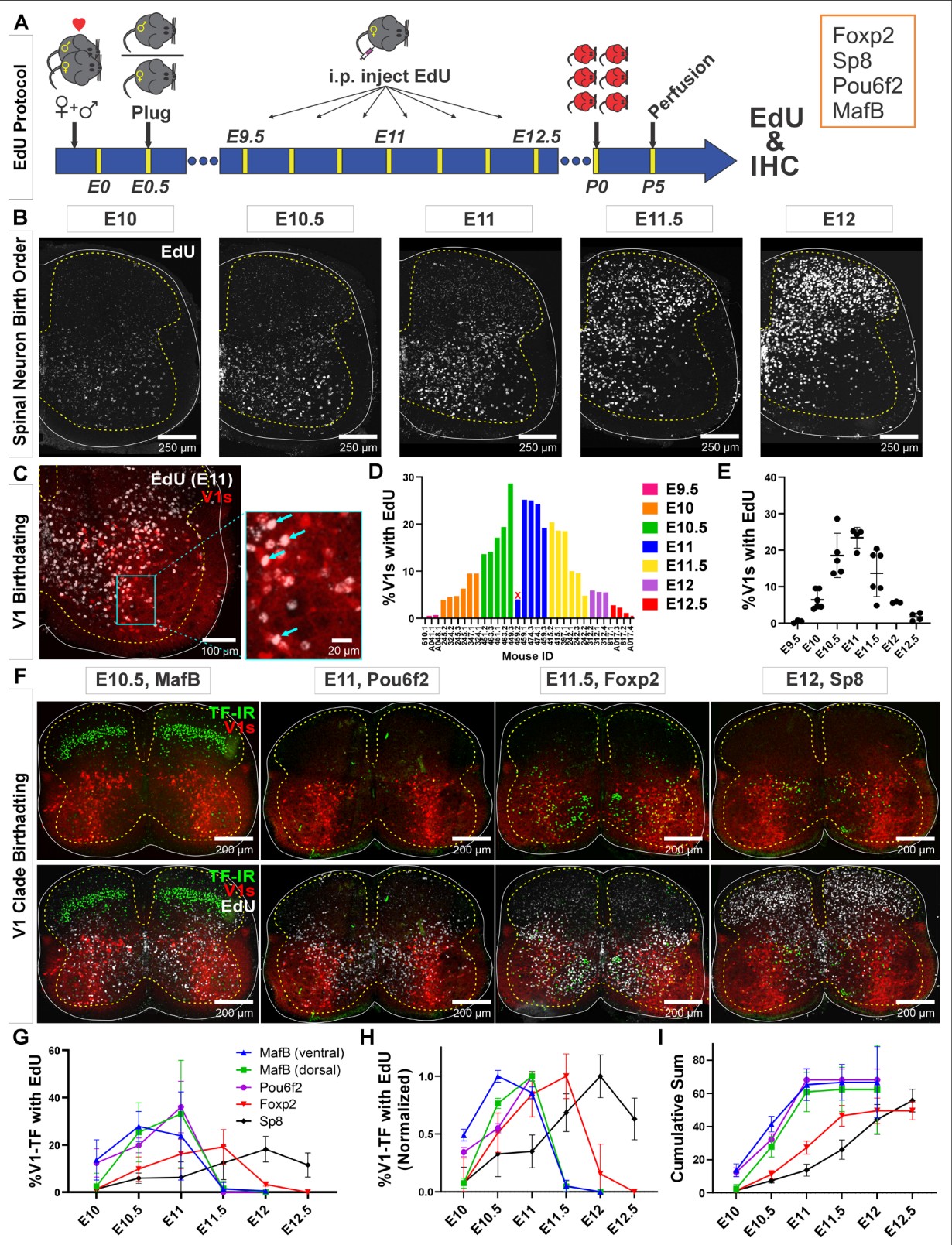

**Figure 1.** Neurogenesis order of V1 clades assayed by 5-ethynyl-2'-deoxyuridine (EdU) birthdating. (**A**) Experimental design. Timed pregnant *En1^cre^*,Ai9 *R26^lsl-tdT^* females were EdU-injected at one of seven time points between E9.5 and E12.5, and the spinal cords harvested at P5. Tissue sections were processed for EdU (Click-iT) and immunostained for representative transcription factors (TFs) of major V1 clades. MafB was used to identify the MafA-Renshaw cell clade by location ('ventral MafB-V1 cells'). (**B**) EdU labeling at different embryonic times. Spinal neurons are born in a ventrolateral to

*Figure 1 continued on next page*

*Figure 1 continued*

dorsomedial sequence. (**C**) Example of E11 EdU labeling in an En1-tdT spinal cord. EdU integrated in the DNA at the time of injection is diluted with subsequent cell divisions. To ensure we sampled V1 cells that incorporated EdU during S-phase after their final division, we only counted V1 cells with nuclei filled by EdU Click-iT reaction (arrows). Partially labeled nuclei (speckles) in the image were not counted. (**D**) Percentages of V1 cells labeled with 'strong' EdU in each mouse. The x-axis indicates individual animals ('<litter number>.<animal number>'). The percentage of EdU-labeled V1 interneurons at each time point was consistent, although we also noted variability between litters and among animals within a litter. One animal (459.2) showed the wrong EdU pattern for its injection age and was discarded (indicated by an X). (**E**) EdU birthdating reveals a peak in V1 neurogenesis around E11 (error bars = SD; each dot represents one animal). At the time points flanking the peak there is a larger amount of variability, suggesting a fast-changing pace in V1 neurogenesis. (**F**) Representative images of TF antibody staining combined with EdU labeling to determine birthdates of defined V1 clades. The time points represented were selected according to the maximal or near-maximal generation of V1 interneurons in each clade. (**G–I**) V1-clade neurogenesis quantification. Graphs represent the average ± SEM calculated from n=3.9 ± 0.3 mice per TF/date (not all TFs were tested in all mice). For each mouse average, we analyzed four ventral horns in Lumbar 4 or 5 segments (further details of sample structure are explained in the results section). (**G**) Percentage of V1s expressing each clade-specific TF labeled with EdU at each embryonic time point. Ventral MafB-V1s (Renshaw cells) and Pou6f2-V1s are mostly born before E11 (dorsal MafB-V1s are a subgroup of Pou6f2-V1s). Foxp2-V1 and Sp8-V1 interneurons have wider windows of neurogenesis, but most are generated after E11. (**H**) Data normalized to the maximum percentage of V1s born in each group showing peak generation in each clade. Ventral MafB-V1s, Pou6f2-V1s, Foxp2-V1s, and Sp8-V1s have progressively later times of peak generation. (**I**) Cum-sum graphs of V1-clades neurogenesis. Between 50% and 68% of all neurons in each V1 clade are labeled across all ages. By E11 nearly all neurons in early clades are generated, while fewer than half of neurons in late clades are.

The online version of this article includes the following source data and figure supplement(s) for figure 1:

**Figure supplement 1.** Foxp2 antibody characterization.

**Figure supplement 1—source data 1.** Raw images of gel corresponding to *Figure 1—figure supplement 1C*.

**Figure supplement 1—source data 2.** Annotated full image of gel corresponding to *Figure 1—figure supplement 1C*.

**Figure supplement 2.** Each antibody was directed against different regions of the mouse MafB protein: aa18–167 for the Sigma antibody (Lot#A31532) and aa100–150 for the Novus antibody (lot#1).

**Figure supplement 2—source data 1.** Raw images of triple probed gel corresponding to *Figure 1—figure supplement 2F*.

**Figure supplement 2—source data 2.** Annotated images of triple probed gel corresponding to *Figure 1—figure supplement 2F*.

**Figure supplement 2—source data 3.** Annotated full image of c-Maf probed gel corresponding to *Figure 1—figure supplement 2F*.

**Figure supplement 3.** MafB-V1s visualized in a MafB-GFP mouse model.

Birthdating divided V1 clades into two groups: early and late born. Most early born V1 cells are EdU-labeled before E11 and include Renshaw cells and Pou6f2-V1s. Most cells from the two other clades (Foxp2-V1s and Sp8-V1s) are late born (after E11) (*Figure 1F–I*). These data derive from 68,562 V1s sampled from 29 animals in 4 ventral horns per animal/TF/EdU time point (average 197.0 V1s per ventral horn). V1 clades differ in birthdate peak times and temporal spread. Ventral MafB-V1 Renshaw cells, Pou6f2-V1s, and dorsal MafB-V1s display early peaks (at or before E11) and narrow spreads in their birthdates. Foxp2-V1s and Sp8-V1s show later peaks and larger spreads (*Figure 1G and H*). Normalizing to their peaks, reveals that ventral MafB-V1 Renshaw cells are generated the earliest (E10.5 peak), followed by Pou6f2-V1s including dorsal MafB-V1s (E11.0 peak), Foxp2-V1s (E11.5 peak), and Sp8-V1s (E12 peak). Around 50% of all V1s are born by E11, including most cells in early born clades and a proportion of Foxp2-V1s and Sp8-V1s (*Figure 1I*). Less than 10% of Foxp2-V1s and Sp8-V1s are generated before E10.5, and almost no ventral MafB-V1s or Pou6f2-V1s (including the dorsal MafB-V1s) are generated after E11 (*Figure 1I*). After E12, only Sp8-V1s are generated. Cumulative graphs show that EdU labeling accounted for 67.3% of ventral (Renshaw) MafB-V1s, 62.4% of Pou6f2-V1s, 68.2% of dorsal MafB-V1s, 49.8% of Foxp2-V1s, and 55.8% of Sp8-V1s. This suggests that our analyses represent half or more of the cells in each of the clades and are representative samples to conclude differences in birthdate based on EdU pulse labeling at seven discrete time points during a 96 hr neurogenesis period.

Next, we examined whether clade-specific V1 interneurons with different birthdates settle in different locations by plotting the positions of V1 interneurons from each clade born at different embryonic times (*Figure 2*). We analyzed E10 to E12 because few V1s incorporated EdU at E9.5 and E12.5 and this diminishes the accuracy of cellular density plot representations. The earliest born MafB-V1s (E10) accumulate ventrally in the 'Renshaw cell area'. Later born MafB-V1s (E10.5 and E11) are a subgroup of Pou6f2-V1s and occupy dorsal positions. Pou6f2-V1s, overall, are born between E10 to E11 and settle dorsally. In contrast, Foxp2-V1 generation spans the entire V1 neurogenesis period and shows differences in location according to birthdate, following a clockwise rotation in positioning

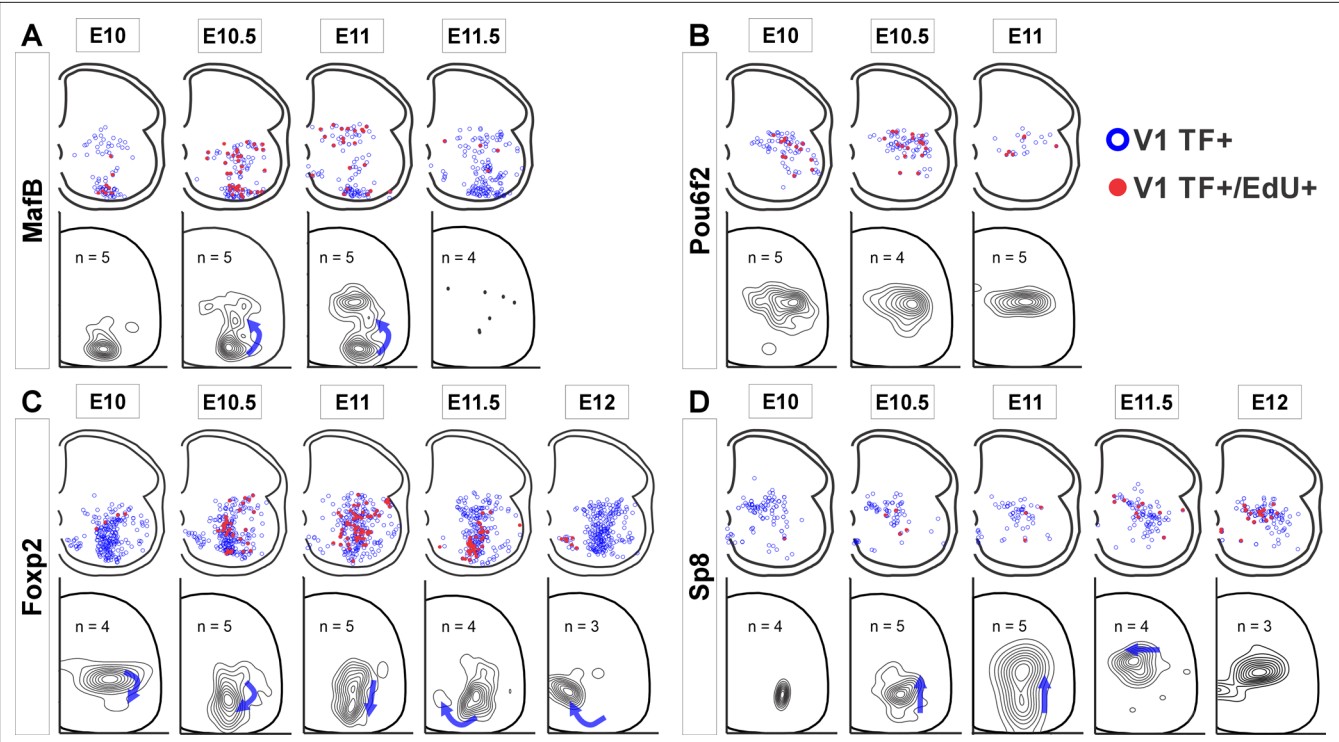

**Figure 2.** V1 interneurons born at different embryonic times settle in different locations. Top row in each panel: blue open dots indicate all V1 interneurons positive for each transcription factor (TF) at P5, and filled red dots indicate those with strong 5-ethynyl-2'-deoxyuridine (EdU). Each plot is from one representative animal injected at each of the indicated embryonic times (four ventral horns from each animal are superimposed in the diagram). Bottom rows: cellular density profiles of V1 interneurons positive for each transcription factor (TF) and with strong EdU. Cellular density profiles represent all cells sampled in all animals EdU-injected at the indicated embryonic times; n indicates the number of animals in each plot (4 ventral horns analyzed per animal). Contour plots are derived from 2D kernel density estimates of interneuron positions; lines encompass 10% increments. Only time points with representative numbers of TF+/EdU+ cells are shown. Blue arrows highlight major directional changes in settling locations for V1s born at each time point with respect to the previous time point. (**A**) MafB-V1s are divided into two groups: ventral Renshaw cell MafB-V1s and dorsal MafB-Pou6f2-V1s with different birthdates. (**B**) Pou6f2-V1s (taken as a whole) are born between E10 and E11.5 and always settle dorsally. (**C**) Foxp2-V1s show variations in location according to birthdate. Cells of increasingly older birthdates settle dorsally, laterally, ventrolaterally, and then ventromedially. (**D**) A few Sp8-V1s are born early and are located ventrally, but the majority are born later and settle dorsally. Directional changes in settling positions according to birthdate differ among V1 clades.

with time: Foxp2-V1s born at E10 occupy dorsal locations while those born between E10.5 and E11.5 settle adjacent to the LMC. Among these, there is a dorsal to ventral progression of Foxp2-V1 cells born from E10.5 to E11. Finally, the few Foxp2-V1s born at E12 are located ventromedially. Sp8-V1 neurons born at different times also show differences in location. Early born Sp8-V1 cells are ventrally located, while later born Sp8-V1 cells are located more dorsally and medially.

In summary, different V1 clades are generated through overlapping windows of neurogenesis, but with distinct peaks which allow classification into early (Renshaw cells and Pou6f2-V1s) and late born clades (most of Foxp2-V1s and Sp8-V1s). Within clades, cells with different birthdates settle at specific positions, perhaps suggesting cellular and functional heterogeneity. This is most evident for the large Foxp2-V1 clade. Birthdate-related settling positions do not follow the general rule of ventrolateral to dorsomedial positions with neurogenesis progression that is evident when considering all spinal neurons. This suggests specific migratory and settling behaviors for specific V1 subgroups being these predictive of cell-specific connectivity, as shown earlier for Renshaw cells (*Benito-Gonzalez and Alvarez, 2012*).

## MafB-V1 genetic labeling reveals two main types in the mature spinal cord

We used two MafB antibodies to identify the Renshaw cell clade because we were unable to reliably label Renshaw cells with MafA antibodies at P5. The two different MafB antibodies differ in

sensitivity and specificity of labeling (*Figure 1—figure supplement 2*), but both consistently labeled ventral MafB-V1 Renshaw cells and dorsal MafB-Pou6f2-V1 cells. For most studies we used the antibody that best labeled Renshaw cells, although this was also the less specific antibody. As explained in *Figure 1—figure supplement 2*, the better labeling with this antibody might be due to cross-reactivity with MafA, a related TF specifically expressed also in Renshaw cells. Neither MafB antibody labeled V1 cells after P5. To further confirm the identify of these cells and study them in older animals, we used a genetic detection strategy by introducing a *Mafb*$^{GFP}$ reporter allele (*Lane et al., 2021Moriguchi et al., 2006*) in *En1*$^{cre/+}$, Ai9 *R26*$^{lsl-tdT}$ mice (*Figure 1—figure supplement 3*). In these animals, GFP is present in many spinal neurons and microglia. Neurons belonging to the V1 group are easily identified by tdT labeling. The two main groups of MafB-V1 interneurons, ventral calbindin-immunoreactive (-IR) Renshaw cells and dorsal Pou6f2-V1s, are easily identified at both P5 and P15 (*Figure 1—figure supplement 3A and B*). At P5, weak GFP is also visible in a few V1 cells located in the middle of the ventral horn that likely belong to the Foxp2-V1 clade; *Mafb* gene expression in Foxp2-V1s cells is probably downregulating at P5 because fluorescence is weak, shows high inter-animal variability, and it is absent at P15. We focused the analyses on the brighter dorsal and ventral MafB-V1 populations defined by boxes of 100 μm dorsoventral width at the level of the central canal (dorsal) or the ventral edge of the gray matter (ventral) (*Figure 1—figure supplement 3B*). MafB-V1s in these two regions together constitute 13.2%±2.8 (mean ± SD) of all V1s at P5 (n=17 mice) with similar representation in dorsal (7.1%±2.4) and ventral (6.0%±1.2) groups (*Figure 1—figure supplement 3C*). These percentages did not change at P15. At P5, 54.2%±19.4 of dorsal and 70.5%±10.1 of ventral MafB-V1s had detectable MafB-IR (*Figure 1—figure supplement 3D*, n=9 mice tested). No MafB protein was detected by P15. However, *Mafb* gene expression is present in adult animals, as reported by GFP (after antibody amplification). Since MafB protein is undetectable this suggests post-translational regulation of MafB protein translation and/or stability.

We next tested V1 clade immunomarkers at P5 (*Figure 1—figure supplement 3E*). Dorsal MafB-V1s expressed Pou6f2 (41.1%±7.4, n=5 mice tested), almost no calbindin (1.5%±1.8, n=4 mice), and no Foxp2 or Sp8 (0%, n=5 mice). Most ventral MafB-V1s expressed calbindin-IR (79.3%±8.8, n=4 mice) while expression of other clade markers was negligible (Pou6f2, 0%; Foxp2, 0.4%±0.9; Sp8. 0.7%±1.5, n=5 mice) (*Figure 1—figure supplement 3E*). We then confirmed birthdates of dorsal (Pou6f2) and ventral (Renshaw) MafB-V1s in the genetic model (*Figure 1—figure supplement 3F*, n=2 mice per EdU injection date). The data agreed with birthdate estimates for dorsal and ventral MafB-immunoreactive V1 neurons, except that we found more ventral MafB-V1s born at E11 using genetic labeling compared to antibody staining. But this difference did not reach statistical significance. In conclusion, we confirmed both MafB-V1 populations and their early birthdates using a genetic model that provided robust and long-lasting labeling in mature spinal cords. These two V1 classes constitute discrete populations; however, while the connectivity of Renshaw cells is well known, the significance of adult dorsal Pou6f2-MafB-V1 interneurons remains unexplored.

## Genetic labeling of the Foxp2-V1 lineage reveals twice the number of neurons relative to postnatal Foxp2 expression

To label the Foxp2-V1 lineage independent of the developmental regulation of Foxp2 expression, we utilized an *En1* and *Foxp2* intersection by generating *En1*$^{cre/+}$, *Foxp2*$^{flpo/+}$, *R26*$^{RCE:dualEGFP/Ai9-lsl-tdT}$ mice. In these mice we expected that neurons expressing *En1* and not *Foxp2* would be labeled with tdT from the R26-Ai9 cre reporter, and that neurons that expressed both *En1* and *Foxp2* would be labeled with EGFP from the R26-RCE:dualEGFP and not with tdT because the Ai9 cassette is flanked by FRT sites that are removed by Flpo. We confirmed tdT or GFP labeling in most V1 interneurons, but a few expressed both fluorescent proteins (*Figure 3A*). We believe this is the consequence of Flpo recombination inefficiencies in the larger FRET-flanked Ai9 cassette compared to the much shorter FRET-stop signal in the RCE:dualEGFP. We thus interpret 'yellow' V1 neurons as cells that express the *Foxp2* gene either for a short period of time, weakly, or both. For mapping purposes, we included 'yellow' cells into the Foxp2-V1 clade since we used only the RCE:dualEGFP reporter in follow-up analyses. The whole population of Foxp2-V1 interneurons (green+yellow) cells represents between 59% and 66% of all V1s (*Figure 3—figure supplement 1*). The distribution of Foxp2-V1s (green) and non-Foxp2 V1s (red, not green) overlapped in the ventral horn (*Figure 3B*) and their proportions were constant across postnatal ages (n=6 ventral horns from 1 mouse at P0, P15, and adult and 36 ventral

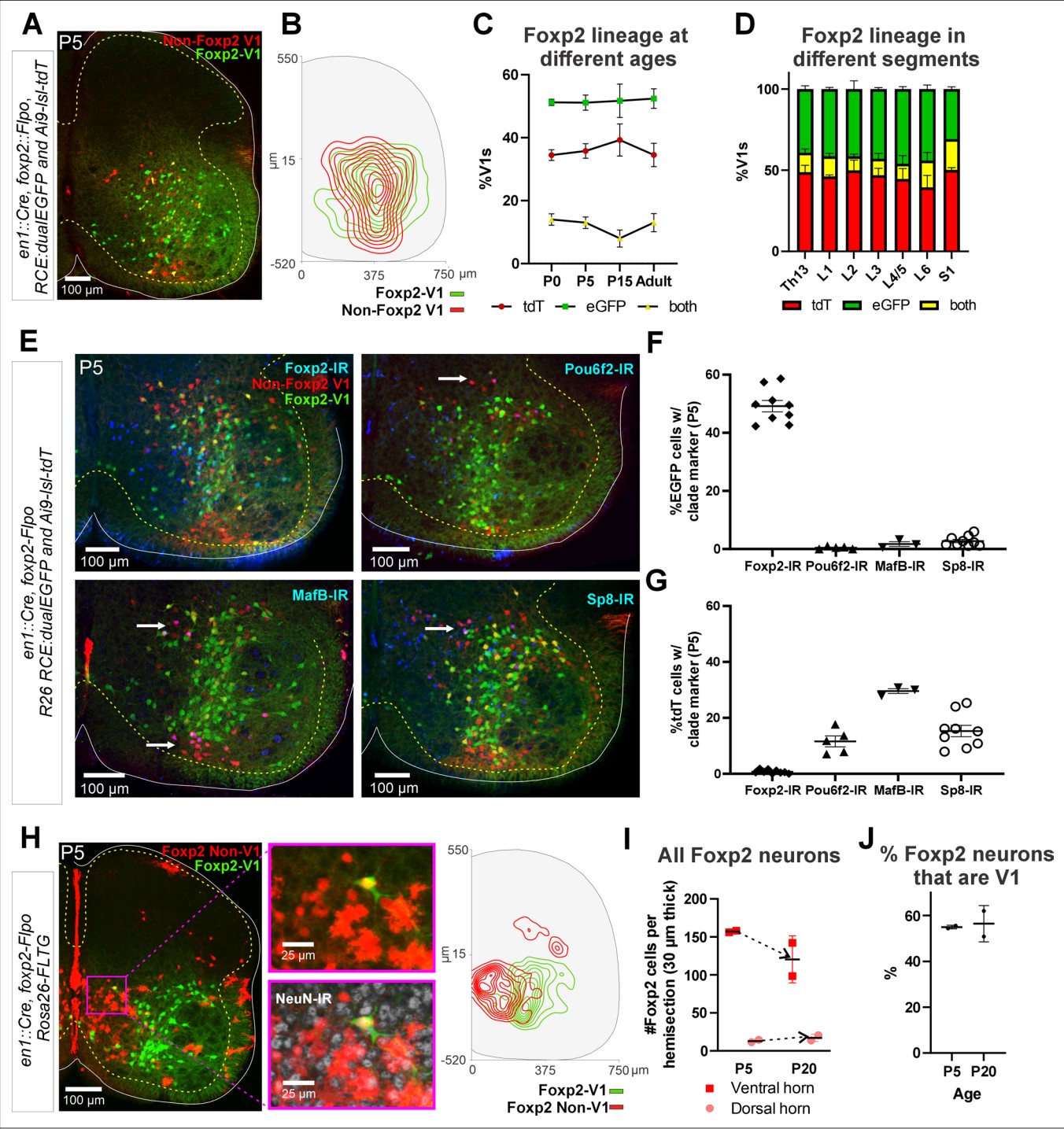

**Figure 3.** Genetic mouse models to label the Foxp2-V1 lineage. (**A**) P5 mouse intersection of *En1^Cre/+* and *Foxp2^Flpo/+* with two reporter alleles (Ai9 *R26 lsl-tdT* and *R26 ^RCE:dualEGFP*). Foxp2-V1s express EGFP (green) and non-Foxp2-V1s tdTomato (red). A few 'yellow cells' correspond with Foxp2-V1 neurons that failed to remove the tdTomato Ai9 reporter. (**B**) Density contours demonstrate high spatial overlap between V1 neurons expressing Foxp2 (EGFP+) or not (tdT+ and EGFP-) (n=6 ventral horns). (**C**) Expression of lineage labels are stable throughout postnatal development, suggesting no additional *Foxp2* gene upregulation in V1s after P0 (n=6 ventral horns per age in one animal, except for P5 in which 6 animals and 36 ventral horns are included; error bars show SEM). (**D**) Lineage labeling in P15 mice is uniform across all spinal cord segments from thoracic 13 to sacral 1 (n=6 ventral horns in each segment from 2 mice; error bars show SEM). (**E**) Foxp2-V1 (EGFP) and non-Foxp2 V1 (tdT) lineage labeling with antibody staining for the transcription factors defining the four major V1 clades: Foxp2, Pou6f2, MafB, and Sp8. The Foxp2-V1 lineage contains all P5 Foxp2-expressing V1s and excludes almost all those expressing the markers of other clades. (**F**) Percentages of Foxp2-V1s (EGFP) expressing the clade markers at P5. Around half of the Foxp2-V1s maintain expression of the Foxp2 protein at P5, and a minimal number of these cells express transcription factors that define other V1 clades.

*Figure 3 continued on next page*

Figure 3 continued

(**G**) Percentages of non-Foxp2 lineage V1s (tdT) expressing the different V1 clade markers. Cells outside of the Foxp2-V1 lineage do not express Foxp2 at P5, and this subset contain V1s from the three other clades (for both plots n=6.5 ± 2.6 mice, 4 ventral horns each; error bars show SEMs). (**H**) FLTG reporter mice reveal lineage-labeled non-V1 Foxp2 cells in the spinal cord. EGFP is expressed in Foxp2-V1s and tdTomato in non-V1 Foxp2 cells. The zoomed images of the highlighted region with and without NeuN-IR demonstrate that non-V1 Foxp2 cells include non-neuronal cell types with astrocyte morphologies. Only neurons (NeuN-IR) are included in the cell density contour plots; non-V1 Foxp2 neurons are located in the medial ventral horn and a few in the deep dorsal horn (n=2 animals at P5, 6 ventral horns in each). (**I**) Most Foxp2-neurons (red, non-V1s or green, V1s) are in the ventral horn and their number in 30-μm-thick L4-5 sections decreases with age as the neuropil matures and expands in size. Dorsal horn Foxp2-neurons maintain their numbers despite the growth of the spinal cord, suggesting de novo postnatal Foxp2 expression in this population. Each point is one animal analyzed through 6 ventral horns (errors bars are SD). (**J**) Percentage of Foxp2-lineage-labeled cells that are Foxp2-V1s remains constant throughout postnatal development (n=2 mice, 6 ventral horns each; error bars show SD).

The online version of this article includes the following figure supplement(s) for figure 3:

**Figure supplement 1.** Postnatal downregulation of Foxp2 expression.

horns from 6 mice at P5; *Figure 3C*) and spinal segments from lower thoracic to sacral level (2 mice at P15 analyzed in n=6 ventral horns per segment and mouse, *Figure 3D*). In each mouse, 51% and 52% of V1 cells were GFP labeled, 34% and 39% were tdT labeled, and 8% and 14% had both labels. These data suggest a consistent population through different ages with no new expression of Foxp2 in V1 cells after birth. Previous data based on protein detection at P0 suggest that only 32–34% of V1s express Foxp2-IR (*Bikoff et al., 2016*). Correspondingly, only around half of genetically identified Foxp2-V1 cells were Foxp2 immunolabeled at P5 in dual-color R26 RCE:dualEGFP/Ai9-lsl-tdT mice (*Figure 3F*, n=2 mice, 3 ventral horns each) or in single-color R26 RCE:dualEGFP mice (*Figure 3— figure supplement 1*, n=3 mice, 4 ventral horns each). The locations of lineage labeled Foxp2-V1 cells with and without Foxp2 immunoreactivity at P5 overlapped (*Figure 3—figure supplement 1*). Lineage labeled Foxp2-V1 cells did not express markers of other V1 clades, suggesting they uniquely represent the Foxp2-V1 clade. Non-Foxp2 tdT-only V1 cells expressed all other clade-specific TFs (Pou6f2, MafB, and Sp8) and consistently lacked Foxp2 immunoreactivity (*Figure 3G*).

## Non-V1 Foxp2 cells distribute to separate regions of the spinal cord

The Foxp2 protein is also expressed in non-V1 cells (*Figure 1F*, *Figure 3—figure supplement 1*). To study this population we used RC-FLTG reporter mice carrying a dual-conditional allele with an FRT-flanked stop and loxP-flanked tdT::STOP preventing transcription of EGFP. Therefore, in *En1*$^{cre/+}$, *Foxp2*$^{flpo/+}$ mice, cells that express only *Foxp2* are labeled with tdT, while additional *En1* expression (Foxp2-V1 clade) results in EGFP fluorescence and removal of tdT. We analyzed two mice at P5 and two at P10, and the sections were counterstained with NeuN antibodies for neuronal confirmation (*Figure 3H*). We found several non-overlapping populations of non-V1 Foxp2 cells. Most are located medially in the ventral horn and can be either neurons or astrocytes (by morphology and lack of NeuN). In addition, cells in the central canal and spinal cord midline were strongly labeled. This distribution suggests Foxp2 is transiently expressed in some progenitors that are different from p1 (no EGFP astrocytes are present). A few non-V1 neurons (NeuN+) are in the deep dorsal horn. The number of Foxp2 neurons per section diminished in the ventral horn from P5 to P20. This is expected because of the decrease in cellular density that occurs as the spinal cord grows and matures. In contrast, dorsal horn Foxp2 neurons slightly increase in number (*Figure 3I*), suggesting some might upregulate Foxp2 postnatally. In summary, there are at least three broad classes of Foxp2 neurons in the spinal cord: (1) medioventral non-V1 neurons that express Foxp2 postnatally and/or at the progenitor stage, sharing labeling with glial cells; (2) dorsal horn non-V1 neurons in which Foxp2 expression increases during postnatal development; (3) V1 neurons that upregulate Foxp2 in embryo and then remain a stable population postnatally (see also *Benito-Gonzalez and Alvarez, 2012*). Overall, lineage-labeled Foxp2-V1 interneurons comprise 55.7%±4.7 (mean ± SD) of all spinal cord Foxp2+ neurons at P5 (n=4 mice, 2 P5 and 2 P20, *Figure 3J*).

## Foxp2-V1 neurons from lower thoracic to sacral segments follow motoneuron numbers in a 2:1 or 3:1 ratio

To gain insights into possible functions of Foxp2-V1 interneurons, we analyzed their localization and numbers across spinal cord segments in the lumbosacral region that govern the lower body and

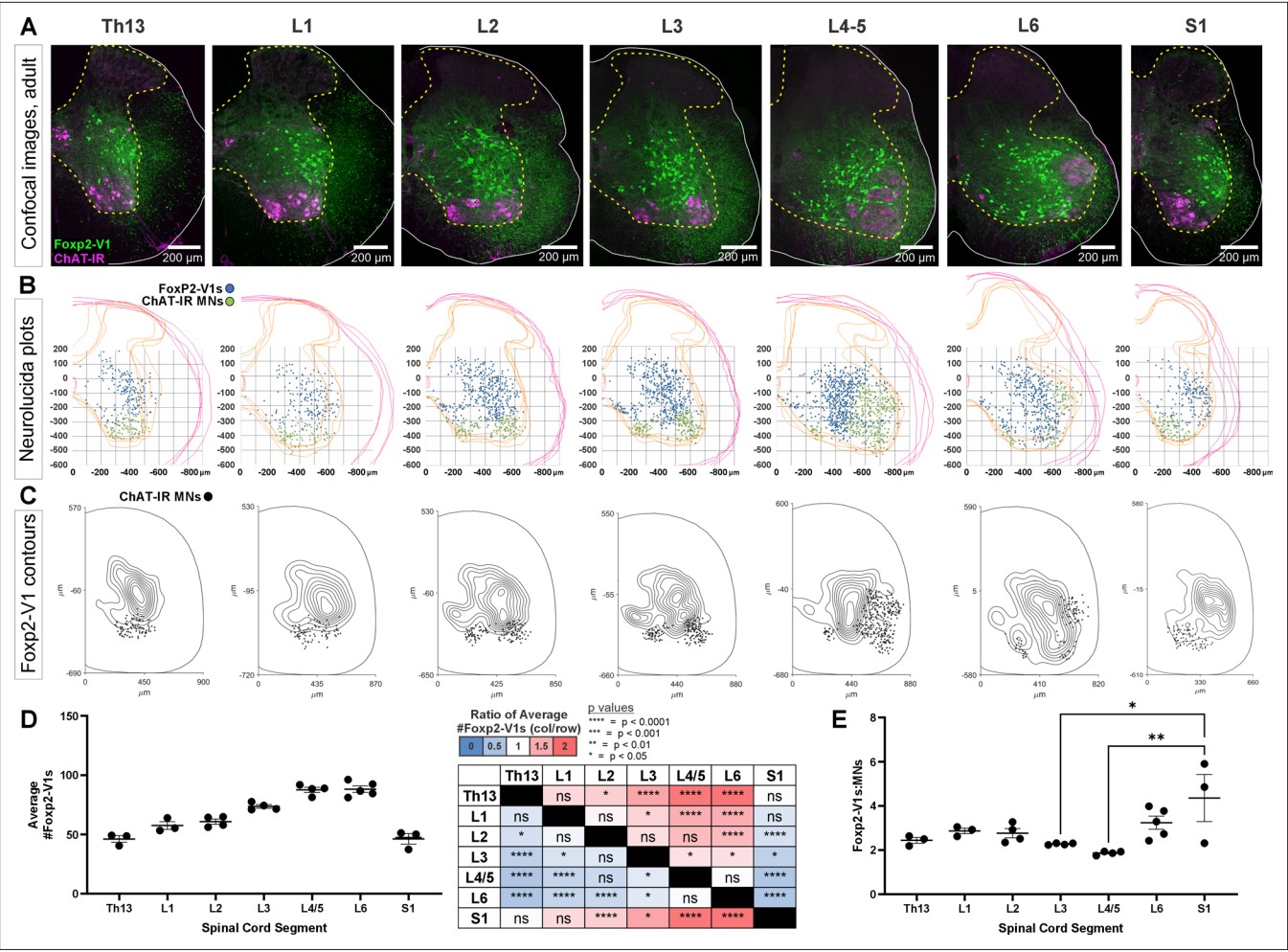

**Figure 4.** Foxp2-V1 interneurons are closely associated with shifting motor columns throughout thoracic, lumbar, and sacral levels of the spinal cord. (**A**) Foxp2-V1 lineage labeling and choline acetyltransferase (ChAT) antibody staining for motoneuron identification in adult mouse spinal cords from thoracic to sacral levels. Foxp2-V1 neurons accumulate at the lateral edge of the ventral spinal cord but their locations shift when the lateral motor column (LMC) expands from L3 to L6. In addition, a distinct group of Foxp2-V1 interneurons is dispersed at ventromedial locations in lumbar segments. (**B**) Plots of Foxp2-V1 and ChAT-IR motoneuron cell body positions in x,y coordinates with 0,0 at the top of the central canal (n=4 ventral horns, 1 representative animal). (**C**) Contour plots of kernel Foxp2-V1 cell density estimations. The highest density of Foxp2-V1 neurons cluster close to LMC motoneurons from L2 to L5 (contours enclose 10% increments, closer lines indicate steeper changes in density). Motoneuron numbers progressively increase from Th13 to L5 and drop in number in L6 and S1. (**D**) Number of Foxp2-V1s per 50-µm-thick section (ventral horn) significantly increases in lower lumbar segments from L3 to L6 compared to S1 (dots represent individual mice; n=3–5 mice in different segments, each mouse estimate is from 6 ventral horns; bars show SEM). One-way ANOVA, post hoc Bonferroni-corrected t-tests are summarized in the right-hand table (further statistical details in **Supplementary file 1a**). (**E**) Ratios of Foxp2-V1 neurons to MNs remain constant at roughly 2.5:1 with no significant changes throughout the lumbar cord. Significance was only found for L3-L5 compared to S1 (*p<0.05; **p<0.01; post hoc Bonferroni tests). High variability in S1 is likely due to the sharp rostro-caudal decrease in motoneuron numbers in S1.

hindlimbs. We examined segments Th13 to S1 in P20 mice (n=5) expressing EGFP in Foxp2-V1 interneurons. This was combined with choline acetyltransferase (ChAT) immunoreactivity to identify the different motor columns. Motoneurons were defined as ChAt-IR neurons located in LIX (**Figure 4A**). Spinal segments were identified by the distribution and size of the somatic lateral, hypaxial, and medial motor columns (LMC, HMC, and MMC, respectively), and presence of autonomic sympathetic (Th13-L2) or parasympathetic (S1) neurons. We did not attempt to distinguish Lumbar 4 from 5 because of their similarity. From these sections, we constructed cell plots for each animal (4 ventral horns per animal/segment, **Figure 4B**) and transformed these into density plots (**Figure 4C**) by combining all cell plots from all animals analyzed in each segment (n=3–5 mice depending on segment). Foxp2-V1 interneurons are located throughout the ventral horn, but many accumulate laterally. In segments

where the LMC expands, Foxp2-V1 interneurons border the LMC medially. Contour density plots indicate that the highest density of Foxp2-V1 interneurons lies adjacent to the LMC, suggesting a close relationship between Foxp2-V1 interneurons and the motoneurons that control limb musculature. This is consistent in segments where the LMC emerges (L2), disappears (L6), or reaches its maximal size (L4/5). Correspondingly, Foxp2-V1 neuron numbers significantly increase in segments innervating limb muscles (L3-L6) compared to segments involved with axial (Th13, S1) and hypaxial muscles (L1) (p<0.0001, one-way ANOVA followed by post hoc Bonferroni t-tests summarized in *Figure 4D* and *Supplementary file 1a*). The limb-innervating LMC is responsible for most of the change in motoneuron numbers across different spinal cord segments. Consistent with the parallel increase in number of V1 interneurons and motoneurons, the ratio of Foxp2-V1 interneurons to motoneurons remained relatively constant from Th13 to L5 (*Figure 4E*). Differences between Th13 and L6 are non-significant (post hoc Bonferroni t-tests, details in *Supplementary file 1a*). The larger ratio at S1 was significant compared to L3 and L4/5 (see *Supplementary file 1a* for details) but the estimated ratios at S1 were highly variable in the three animals studied. This is probably best explained by the rapid diminution of motoneuron numbers in S1 depending on the exact section level. Cell plots in segments lacking LMC limb motoneurons show that most Foxp2-V1 interneurons are located dorsally and distal to the motor pools, with lower density close to MMC motoneurons. Foxp2-V1 interneurons located further away from the motor pools might have roles other than the direct modulation of motoneuron firing. Finally, a sparse group of Foxp2-V1 interneurons is dispersed in the medial ventral horn in all segments. They correspond to the latest born subgroup (and they have a different genetic make-up, see below), suggesting a unique identity.

## Foxp2-V1 synapses target LMC, HMC, and MMC motoneurons, but not preganglionic autonomic neurons

Next, we examined Foxp2-V1 synapses on motoneurons and compared them to other V1 groups at P20: a time point after the critical window of synapse proliferation and pruning for V1 interneurons (*Mentis et al., 2006*; *Siembab et al., 2010*). We examined three mice in which Foxp2-V1 axons express EGFP, and non-Foxp2-V1 axons tdT. Yellow axons were included in the EGFP/Foxp2+ group. We also analyzed two mice in which all V1 axons express tdT through the Ai9 reporter. In these animals we immunostained axons with calbindin antibodies to identify synapses from Renshaw cells. Sections were further immunostained using antibodies against ChAT (to identify motoneuron cell bodies in LIX) and against the vesicular GABA/glycine amino acid transporter (VGAT) to reveal presynaptic vesicle accumulations in genetically labeled axon varicosities. Motoneurons were sampled in different motor columns from Th13 to S1 segments (*Figure 5A and B*) and examined for synaptic contacts from Foxp2-V1, non-Foxp2-V1, all V1, and Renshaw-V1 axons (*Figure 5C1 and C2*). We used rigorous criteria to estimate synaptic contact densities on 3D reconstructions of cell body surfaces (*Figure 5C3*). We calculated overall V1 synaptic densities (green plus red axons in EGFP/tdT dual-color mice, and all red axons in tdT single-color mice) on motoneurons from the following motor columns and segments: HMC motoneurons in the Th13 segment (ventral body musculature); LMC motoneurons in L1/2 (hip flexors), L4/5 (divided into dorsal and ventral pools, innervating distal and proximal leg muscles, respectively) and the dorsal L6 pool (intrinsic foot muscles); MMC neurons in segments Th13, L1/2, L3/4, and S1 (innervating axial trunk musculature and the tail at sacral levels); and finally, preganglionic autonomic cells (PGC), sympathetic at Th13 and L1/2 and parasympathetic at S1. We analyzed 4–9 motoneurons per animal. Initially, we kept the data separated by mouse identity to check for possible differences due to mouse and/or genetics (*Figure 5D*). A mixed-effects nested ANOVA revealed significant differences in V1 synapse density over different types of motoneurons (p<0.0001), but no influence of mouse or genetics (statistics details in *Supplementary file 1b* and *Figure 5D* table). Post hoc Bonferroni t-tests demonstrated that HMC and lower lumbar LMC motoneurons receive significantly more V1 synapses than MMC motoneurons, while LMC motoneurons in L1/2 and L6 had V1 synaptic densities not significantly different to MMC motoneurons. PGC neurons received very low densities of V1 input, significantly lower than LMC or MMC motoneurons.

Next, we examined possible differences between Foxp2-V1 and non-Foxp2-V1 neurons in motoneuron targeting (*Figure 5E*, top graph). In this case we pooled all motoneurons from 3 mice (n=6–16 motoneurons per motor column/segment) and found significant differences according to motoneuron identity (p<0.0001), type of V1 axon (p=0.0107), and their interaction (p<0.0001) (two-way ANOVA,

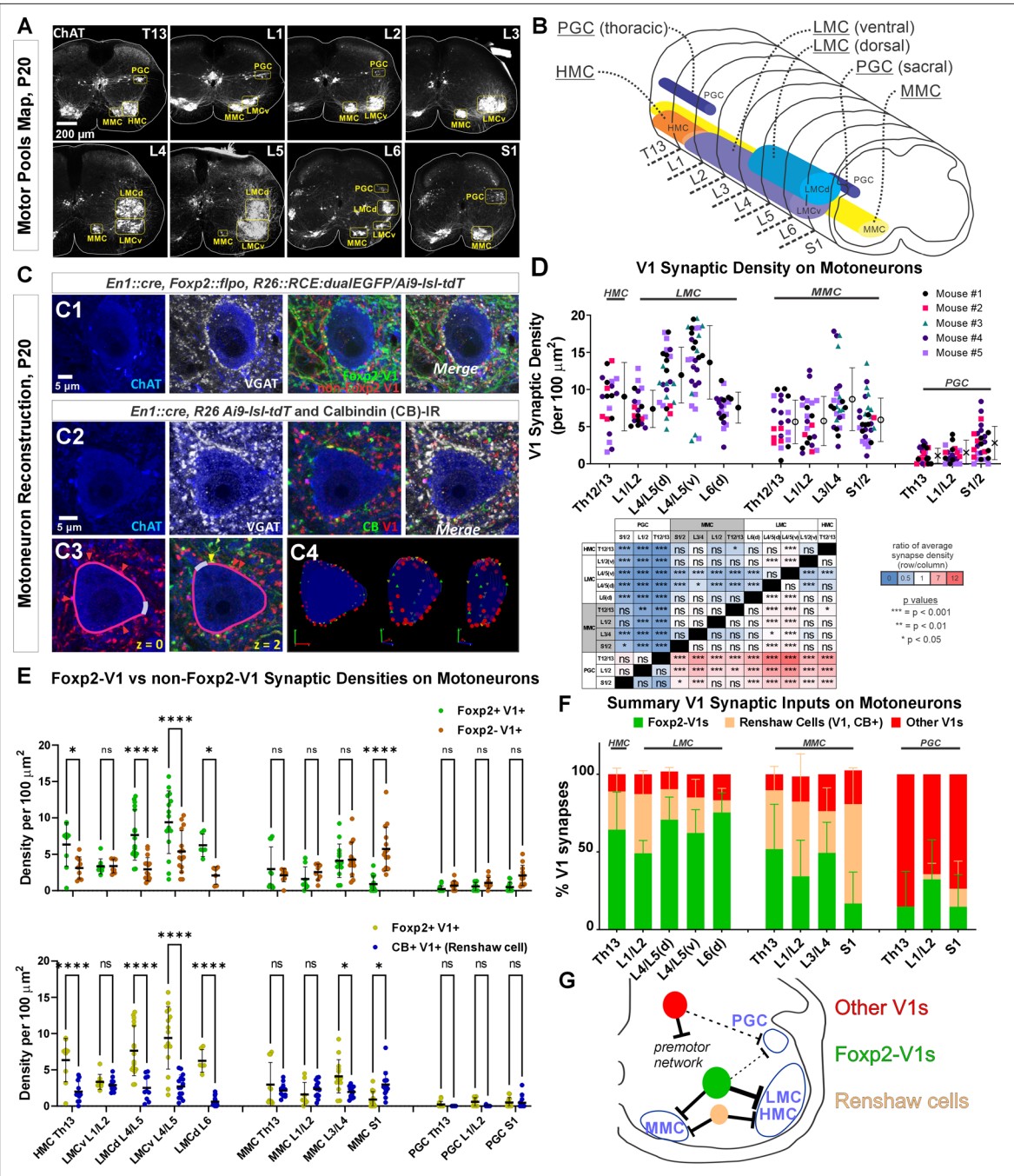

**Figure 5.** Limb and axial motoneurons are densely innervated by Foxp2-V1s and Renshaw cells. (**A**) Motor column identification from lower thoracic to upper sacral spinal cord in P20 mice following labeling with choline acetyltransferase (ChAT) antibodies: PGC = preganglionic cell column; MMC = medial motor column; HMC = hypaxial motor column; LMCd/v=lateral motor column (dorsal/ventral). (**B**) Schematic representation of the rostro-caudal span of each motor column in the spinal segments studied. (**C**) Synapse quantification. Axons of Foxp2 and non-Foxp2 V1 interneurons were respectively labeled with EGFP and tdT in *En1^Cre^, Foxp2^flpo^, R26 ^CE:dualGFP/Ai9-lsl-tdT^* mice. In *en1^cre^, Ai9 R26^lsl-tdT^* mice we identified V1-Renshaw cell axons using calbindin antibodies. Synaptic locations were labeled with VGAT antibodies and the postsynaptic motoneurons with ChAT antibodies. Synapse densities were analyzed in a ribbon of membrane at mid-cell body level (7 optical planes, 1 µm z-step). C1, Single optical plane of an L4/5 LMCv motoneuron surrounded by genetically labeled Foxp2-V1 and non-Foxp2-V1 axons. Inhibitory synapses on ChAT-IR motoneurons are VGAT+. C2, Single optical image of an L4/5 LMCv ChAT-IR motoneuron receiving synapses from V1 Renshaw cells (genetically labeled V1 axons with calbindin-IR and VGAT). C3–4, Method for estimating synapse densities on motoneuron cell bodies using C2 as example. C3, V1-VGAT (red arrowheads) and V1-CB-VGAT synapses (yellow arrowhead) are marked (VGAT-IR is not shown for clarity), and the cell body contour annotated with regions corresponding to dendrite exits. This process was repeated in seven consecutive mid-cell body optical planes (cross-sections with well-defined nucleus and nucleolus). C4, A membrane surface slab is reconstructed in 3D (two different rotations shown). The surface area corresponding to dendrite exits is subtracted from

*Figure 5 continued on next page*

*Figure 5 continued*

the total surface area of the slab to calculate the available surface area on the motoneuron cell body. V1-VGAT synapses (red), V1-CB-VGAT synapses (yellow), and CB-VGAT synapses (green) are marked. A similar process was followed for calculating Foxp2-V1 synapse density. (**D**) Quantification of total V1-VGAT synapse densities on motoneuron cell bodies in different motor columns (n=21–30 motoneurons per motor column, n=5 animals with 4–9 motoneurons per animal per motor column). Each data point is one motoneuron color-coded by mouse origin. Average synaptic densities ± SD indicated to the right of scatter plots. A nested ANOVA found significant differences among motor column/segments (p<0.0001) with no inter-animal variability (p=0.4768). The table summarizes all post hoc pairwise comparisons for average V1 synaptic densities of each motor column and segment (Bonferroni-corrected t-tests) (further statistical details are found in **Supplementary file 1b**). Colors indicate increased (>1, red) or decreased ratios (<1, blue) of column motoneurons vs row motoneurons. PGC neurons receive significantly fewer V1 synapses than MMC or LMC motoneurons. The LMC (ventral and dorsal) in lower lumbar (L4/L5) had significantly more V1 contacts than MMC motoneurons or L6 dorsal LMC. (**E**) Comparison of synaptic densities from Foxp2-V1 and non-Foxp2-V1 neurons (top) or Renshaw cells (bottom). All motoneurons sampled in 2–3 animals for each comparison were pooled together. Densities of V1-VGAT synapses from Foxp2-V1s, non-Foxp2 V1s, or calbindin (CB)+ V1s (Renshaw cells) (n=6–17 motoneurons sampled per motor column/segments, average = 12.1 ± 2.9 SD) were compared using a two-way ANOVA for axon type vs motor column and segment. Foxp2-V1 vs non-Foxp2-V1 synapses: significant differences in density were found for type of synapse (p=0.001), motor column location (p<0.0001), and their interaction (p<0.0001). Significant differences after post hoc Bonferroni tests are indicated (*p<0.05; ****p<0.0001). In general, synapses from Foxp2-V1 axons have higher density than non-Foxp2-V1 axons on HMC and LMC columns at all spinal segments except for L1/L2 LMC. MMC motoneurons receive similar synaptic densities from both types of V1 axons, except at the sacral level in which non-Foxp2 V1 synapses predominate. PGC neurons receive very low densities of V1 axons and there are no significant differences between either type in any region. Foxp2-V1 vs CB+ V1 synapses: significant density differences were found for type of synapse (p<0.0001), motor column location (p<0.0001), and their interaction (p<0.0001). Significant differences between Foxp2-V1 and CB+ V1 synapses after post hoc Bonferroni tests are indicated (*p<0.05; ****p<0.0001). Synapses from Foxp2-V1 axons have higher density than CB+ V1 axons in HMC and LMC columns at all spinal segments except for L1/L2 LMC. MMC motoneurons receive similar synaptic densities from both types of V1 axons in upper lumbar regions, but Foxp2-V1 synapse predominate in lower lumbar. In S1 the density of CB+/V1 synapses is significantly higher. The low synaptic densities estimated in PGC neurons for Foxp2-V1s and CB+ V1s are not significantly different. Further details of statistical comparisons are in **Supplementary file 1c and d**. (**F**) Comparing the numbers of Foxp2 and CB+ (Renshaw) V1 synapses to the total number of V1 synapses, we estimated their respective percentages. From these estimates we calculated that the remainder belongs to non-Foxp2 and non-CB+ Renshaw cells. The large majority of V1 synapses on the cell bodies of LMC, HMC, and MMC motoneurons are either from Renshaw cells or Foxp2-V1s. (**G**) Summary diagram of major V1 clade connectivity to motoneuron cell bodies. Foxp2-V1s and Renshaw cells form the majority of inhibitory V1 contacts on LMC and HMC motoneurons, with slightly higher density from Foxp2-V1s. The MMC receives roughly equal portions of V1 contacts from Foxp2-V1s and Renshaw cells. V1s provide only sparse inhibition on preganglionic sympathetic neurons and most originate in V1 clades other than Renshaw cells and Foxp2-V1s.

statistics details in **Supplementary file 1c**). This was followed by pairwise comparisons of synaptic densities according to the type of V1 axon for each motoneuron type (**Figure 5E**, top graph). HMC and LMC motoneurons receive significantly higher synaptic densities of Foxp2-V1 axons compared to non-Foxp2-V1 axons in all segments, except for L1/2. The synaptic densities of both types of axons are not significantly different in MMC motoneurons, except for S1 MMC motoneurons that received significantly higher density of non-Foxp2-V1 synapses. We conclude that motoneurons controlling the hindlimb receive more synapses from Foxp2-V1 interneurons, while Foxp2-V1 and non-Foxp2-V1 interneurons equally contact cell bodies of motoneurons controlling axial musculature. Synapses on PGC neurons were always at low density and highly variable, with most originating from non-Foxp2 V1 interneurons.

We then compared synapse densities from Foxp2-V1 interneurons to those from Renshaw cells. Renshaw cell axons were identified by the presence of calbindin-IR in tdT V1 axons (in this analysis we pooled 10–17 motoneurons from two mice). Like above, we found significant differences according to motoneuron identity (p<0.0001), type of axon (p<0.0001), and their interaction (p<0.0001) (two-way ANOVA, statistics details in **Supplementary file 1d**). Post hoc pairwise comparisons revealed that Renshaw cell synapses occurred at significantly lower densities compared to Foxp2-V1s in all LMC motor groups, except for L1/2 (**Figure 5E**, lower graph). MMC motoneurons showed similar densities of Renshaw cell and Foxp2-V1 synapses in Th13 and L1/2, higher density of Foxp2-V1 synapses in L3/4, and much higher density of calbindin+ V1 synapses in S1. The identity of calbindin+ V1 axons in S1 is unclear because of the higher numbers of non-Renshaw calbindin-IR V1 interneurons in sacral segments (i.e. calbindin+ V1 interneurons not contacted by motor axons). Further confirmation of Renshaw cell identity for calbindin+ V1s in S1 is required. PGC neurons received almost no calbindin+ V1 axons (**Figure 5E**, bottom graph).

After calculating synaptic densities originating from Foxp2-V1 and Renshaw cell axons, we estimated the remaining V1 synapses. Plotting the percent contributions of three V1 synapse categories to all V1 synapses shows that most synapses on the cell bodies of HMC, LMC, and MMC motoneurons

originate from either Foxp2-V1s or Renshaw cells (*Figure 5F*). PGC neurons mostly receive inputs from V1s that are non-Foxp2 and non-Renshaw cells, but the frequency of these synapses is very low and highly variable. The data suggest that the Foxp2-V1 clade is a major source of inhibitory inputs to motoneuron cell bodies where they likely strongly modulate motoneuron firing. Moreover, Foxp2-V1 synapses preferentially target limb motoneurons. Foxp2-V1 synapses on motoneurons likely originate from those Foxp2 V1 interneurons clustered spatially close to the LMC. Accordingly, motoneurons in spinal segments with Foxp2-V1 interneurons located further dorsally from motor pools (e.g. S1) receive a relatively lower density of synapses from Foxp2-V1 interneurons. Conversely, Pou6f2, Sp8, and other possible V1 clades either do not target motoneurons directly, or they target them sparsely or on distal dendrites. This suggests functional differences among V1 clades in their strength modulating motoneuron firing directly. V1-to-motoneuron soma connectivity is summarized in *Figure 5G*, highlighting the preferred targets of each V1 clade.

## Foxp2-V1 interneurons clustered near LMC motoneurons are genetically distinct

Birthdates, spatial organization, and synapse densities on different motor columns all suggest that Foxp2-V1 interneurons are heterogeneous, and that a laterally located group close to the LMC might modulate the output of limb motoneurons. To identify potential genetic differences among Foxp2-V1 interneurons, we selected for further study two TFs previously found highly enriched in Foxp2-V1 interneurons: Orthopedia homeobox (Otp) and Foxp4 (*Bikoff et al., 2016*). We used antibodies to reveal V1 interneurons expressing Otp and/or Foxp4 in the P5 spinal cord in Lumbar 4 and 5 segments. We first used two dual-color animals with Foxp2 and non-Foxp2 V1 interneurons labeled with EGFP and tdT, respectively. Both Otp and Foxp4 are almost exclusively expressed by Foxp2-V1 interneurons at P5 with negligible expression in non-Foxp2 V1 interneurons (*Figure 6A*). Otp was expressed in around 50% of lineage-labeled Foxp2-V1 interneurons, and Foxp4 in 20%. To examine the relationship between these groups with V1 cells that retain Foxp2 expression at P5, we generated different combinations of paired immunolabelings for Otp, Foxp4, and Foxp2 (*Figure 6B*) in three mice with EGFP expression in Foxp2-V1 interneurons. We constructed cell density contours and calculated the percentages of lineage-labeled Foxp2-V1 cells (EGFP) expressing different combinations of Otp, Foxp4, and Foxp2 (*Figure 6C*). Each TF combination is found in significantly different percentages of cells in the Foxp2-V1 lineage (one-way ANOVA followed by post hoc Bonferroni-corrected t-tests, statistical details in *Supplementary file 1e, f, and g*). To summarize the most salient results, a large group of Foxp2-V1 cells that co-expresses Otp and Foxp2 at P5 (44% of Foxp2-V1 cells) is localized close to the LMC. It includes a smaller subgroup located more ventrally that also expresses Foxp4 (23% of Foxp2-V1 cells). Foxp4-IR/Foxp2-V1 interneurons always co-localized with Otp-IR, indicating they are a subpopulation of the Otp group. Lineage-labeled Foxp2-V1 cells expressing only Foxp2 preferentially occupy a ventro-medial location typical of late born Foxp2-V1 cells (*Figure 2C*), or a dorsal location typical of the earliest born Foxp2-V1 cells.

The birthdates of genetically labeled Foxp2-V1 (EGFP) and non-Foxp2-V1 cells (tdT) (*Figure 6D*) parallel the early (most non-Foxp2 V1s) and late (Foxp2 V1s) times described previously (*Figure 1G and H*). Like Foxp2-IR V1 cells, most cells in the genetically labeled Foxp2-V1 lineage were born between E10.5 and E12.0, with few born at E10.0 or E12.5 (n=1–2 mice per time point). The peak of neurogenesis for the lineage-labeled population occurred at E11.0; i.e., 12 hr earlier than the peak of neurogenesis for V1 cells expressing Foxp2 at P5. This can be explained by a proportion of the earliest born cells in the Foxp2-V1 clade downregulating Foxp2 expression by P5. Next, we used Otp and Foxp4 to subdivide lineage-labeled lateral Foxp2-V1 cells (*Figure 6E*). To increase sample size, we pooled spinal cords from mice in which we genetically labeled Foxp2-V1s (EGFP) and non-Foxp2-V1s (tdT) together with mice having all V1s lineage-labeled with tdT (n=4 at E10, 5 at E10.5, 4 at E11, 3 at E11.5, and 2 at E12.5). Pooling data from both genetic models is justified by the above results showing that V1 cells expressing Otp or Foxp4 at P5 are all contained within the Foxp2-V1 lineage. The neurogenesis curves of V1 cells expressing Otp, Foxp4, and Foxp2 at P5 largely overlapped (*Figure 6F*). The locations of Otp and Foxp4 expressing V1s generated between E10.5 and E11.5 were lateral for Otp and ventral for Foxp4. The location of Foxp4 cells generated at E10.5, E11.0, and E11.5 did not change significantly, but the location of Otp cells generated at E11.5 shifted ventrally

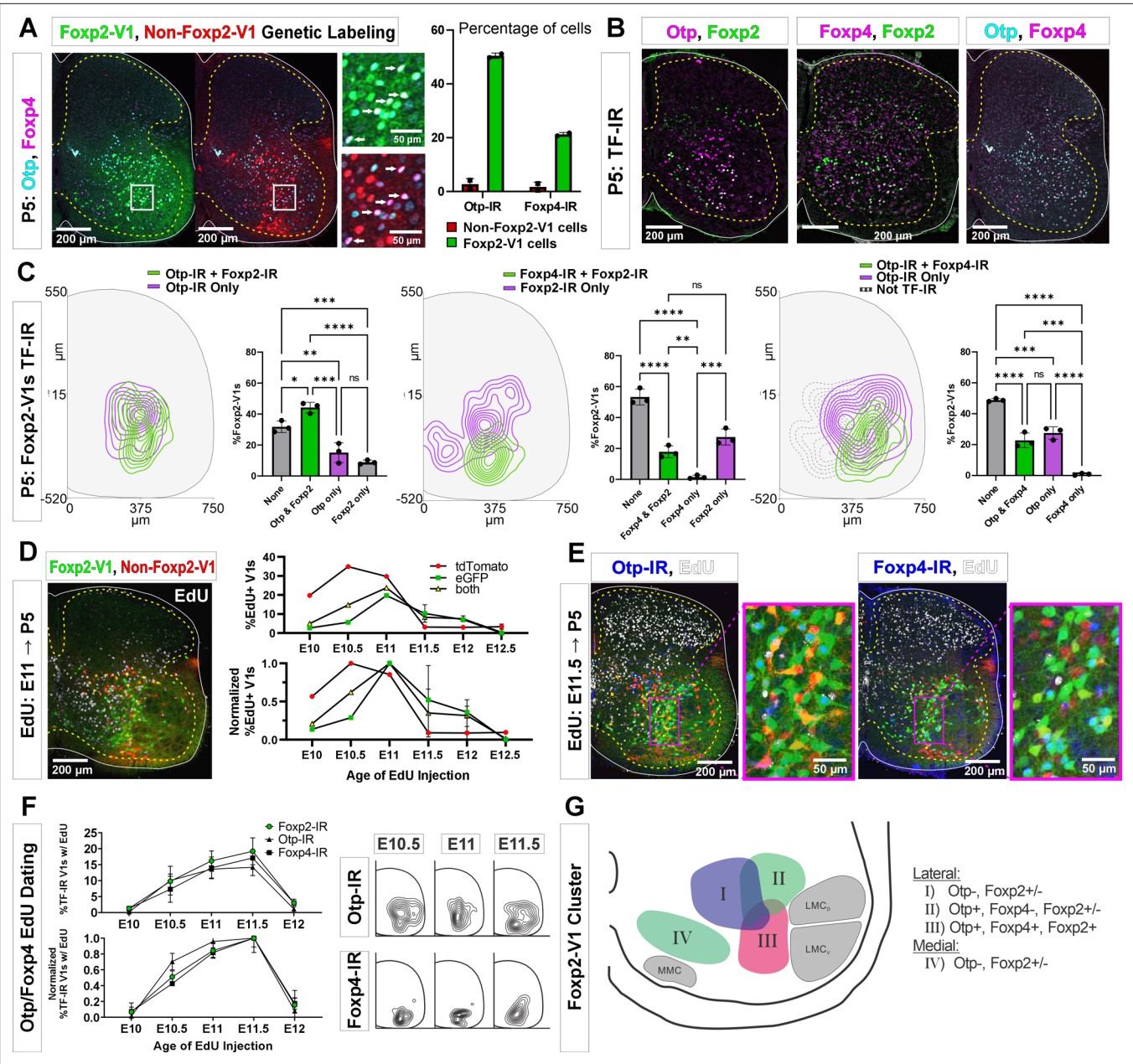

**Figure 6.** Subgroups of Foxp2-V1 interneurons defined by transcription factor (TF) expression at P5 and birthdate. (**A**) Otp (blue) and Foxp4 (white) expression in lineage-labeled Foxp2-V1s (EGFP, green) and non-Foxp2-V1s (tdT, red). The boxed area is shown at higher magnification with different color combinations for clarity. It shows that Foxp4-IR cells in the Foxp2-V1 population always expressed Otp (arrows). Quantification: 49.8–51.2% of Foxp2-V1s express Otp and 20.7–21.7% Foxp4 (n=2 mice each examined in 3 ventral horns in L4/5). Very few non-Foxp2-V1 cells express either TF (Otp: 1.3–4.2% and Foxp2: 0.5–2.9%). (**B**) Images of P5 spinal cords containing Foxp2-V1 lineage labeling (EGFP, omitted for clarity) and double immunolabeled for Otp/Foxp2, Foxp4/Foxp2, and Otp/Foxp4. (**C**) Quantification of Foxp2-V1 interneurons with different combinations of TF expression at P5. For each combination, the left panels show cell distributions, and the right graphs the percentage of Foxp2-V1s with each combination (n=3 mice each analyzed in 6 ventral horns). The data was analyzed with one-way ANOVAs followed by Bonferroni-corrected pairwise comparisons (statistical details in ***Supplementary file 1e, f, and g***. ****p<0.0001; ***p<0.001; **p<0.01; *p<0.05). There are five main groups defined by TF expression patterns: Otp-Foxp2 (44%), Otp-Foxp2-Foxp4 (23%), Otp only (15%), Foxp2 only (9%), and no TF labeling (32% by subtracting all other groups). Some groups associate with specific locations: Otp-Foxp2 cells and the smaller proportion of Otp-only cells are located laterally; Otp-Foxp4-Foxp2 cells are located lateroventrally; medial cells either contain only Foxp2 or nothing; some dorsal cells are either Foxp2 only or do not express any of these TFs. (**D**) Example image of 5-ethynyl-2'-deoxyuridine (EdU) birthdating in *En1^Cre, Foxp2^flpo, R26 ^RCE:dualGFP/Ai9 lsl-tdT* dual-color mice pulse labeled at E11. Foxp2-V1s (green and yellow cells) are born between E10 and E12, with peak birthdate around E11 and after non-Foxp2 V1s (tdTomato only). The lower graph's data is normalized to highlight the time of peak neurogenesis for each population (n=15.0 ± 4.1 ventral horns from one mouse per time point from

*Figure 6 continued on next page*

*Figure 6 continued*

E10 to E11 and 2 mice per time from E11.5 to E12.5; total 9 mice; bars show SEM). There is no difference between green (EGFP only) and the smaller population of yellow cells (EGFP and tdTomato). (**E**) EdU labeling in the Foxp2-V1 dual-color genetic model combined with Otp and Foxp4 antibody staining. (**F**) V1s expressing Foxp2, Otp, and Foxp4 at P5 are mostly born between E10.5 and E11.5 with neurogenesis time courses largely overlapping. The normalized plot indicates that peak neurogenesis for all three populations occurs at E11.5, although a marginally higher number of Otp V1 cells are born earlier (n=4 ventral horns from 3.27±1.34 mice per time point; error bars show SEM). Contour plots to the right show settling locations of Otp and Foxp4-IR populations born at each time point. (**G**) Schematic of the L4-5 ventral horn summarizing Foxp2-V1 subgroups according to location and combinatorial expression of Otp, Foxp4, and Foxp2 at P5. LMC$_D$: dorsal lateral motor column; LMC$_V$: ventral lateral motor column; MMC: medial motor column.

compared to those born at E10.5 or E11.0 (*Figure 6F*). We interpret this result as suggesting a shift in the balance toward generation of ventral Foxp4-Otp Foxp2-V1 interneurons at later times.

The results suggest at least four groups of Foxp2-V1 cells according to location, TF expression, and birthdate (summarized in *Figure 6G*). Group I is located dorsomedially, lacks Otp, and has variable expression of Foxp2 at P5. Many are likely generated during early Foxp2-V1 neurogenesis (before E10.5). Groups II and III are laterally located and express Otp with or without Foxp2 (group II) or express Otp, Foxp4, and Foxp2 together (group III). These two groups together represent the largest class of Foxp2-V1 cells in L4/5 and are generated during a 24 hr period from E10.5 to E11.5 with a slight shift in the balance of group II vs III cells at later times of neurogenesis. Finally, ventromedial Foxp2-V1 cells (group IV) are generated very late (after E12: see *Figure 2C*) and lack expression of Otp or Foxp4, but many retain expression of Foxp2 at P5. We failed to identify in this ventrome-dial Foxp2-V1 group expression of late born markers, like NeuroD2 and Prox1 (*Delile et al., 2019*; *Osseward et al., 2021*), despite the presence of many other neurons positive for these TFs in the vicinity.

## Otp-expressing Foxp2-V1 cells receive proprioceptive synapses

Earlier studies reported that Foxp2-V1 interneurons include a large class of proprioceptive interneu-rons. Some of these could represent reciprocal Ia inhibitory interneurons (IaINs) because they receive convergent synapses from excitatory proprioceptive afferents (VGLUT1+) and inhibitory Renshaw cells (calbindin+ V1 axons) (*Benito-Gonzalez and Alvarez, 2012*). Likely candidates are Otp Foxp2-V1 interneurons in groups II and III because their localization matches that of electrophysiologically iden-tified IaINs in the cat (*Jankowska and Lindström, 1972*; *Alvarez et al., 1997*). To examine this, we first analyzed the types of Foxp2-V1 interneurons receiving proprioceptive inputs. The spinal cords of two mice at P5 (to preserve TF expression) containing Foxp2-V1 lineage-labeled cells (EGFP) were dual or triple immunolabeled for VGLUT1 and Otp and/or Foxp2 (*Figure 7A*). VGLUT1 synapses in the ventral horn at P5 originate solely from parvalbumin+ proprioceptors, the majority being Ia affer-ents (*Alvarez et al., 2004*). We analyzed six ventral horns at L4/L5 in each animal, tiling the whole ventral region containing Foxp2-V1 cells using high-magnification confocal microscopy (×60). Cells were categorized as receiving no synapses (I in *Figure 7A*, ×60 inlay), or low/medium and high density of VGLUT1 synapses (respectively, II and III in *Figure 7A*, ×60 inlay). We also noted whether these synapses were located proximally (on cell body and primary dendrites) or more distally. In general, cells with proximal VGLUT1+ synapses had higher densities than those with only distal synapses. Overall, we found that 63.0% and 74.3% of Foxp2-V1 interneurons received VGLUT1 synapses in each animal respectively (*Figure 7C*). Foxp2-V1 cells with no VGLUT1 synapses were found throughout the ventral horn, but those receiving VGLUT1 synapses had a bias toward lateral positioning (*Figure 7B*).

VGLUT1 synapses preferentially targeted genetically labeled Foxp2-V1 interneurons expressing Otp and Foxp2 at P5 (*Figure 7C and D*). On average (n=2 mice, 12 ventral horns and 1116 Foxp2-V1 cells analyzed), 42.5% of Foxp2-V1 interneurons had VGLUT1 contacts and were Otp+, while 22.1% were Otp(-). Thus, 65.8% of interneurons receiving VGLUT1 synapses in the Foxp2-V1 lineage express Otp. Only 9.3% of Foxp2-V1 interneurons that were Otp+ lacked VGLUT1 synapses (*Figure 7D*). Thus, VGLUT1 synapses contacted 82.0% of Otp+ Foxp2 V1 interneurons and 64.9% received these synapses proximally (cell body and primary dendrites) in addition to also having VGLUT1 synapses on more distal dendrites. Sections immunolabeled for Foxp2 revealed 38.2% of lineage-labeled Foxp2-V1 interneurons with VGLUT1 contacts, expressed Foxp2 while 30.4% were Foxp2(-). Only 14.0% of Foxp2-V1 interneurons retaining Foxp2 expression at P5 lacked VGLUT1 synapses (*Figure 7D*). This

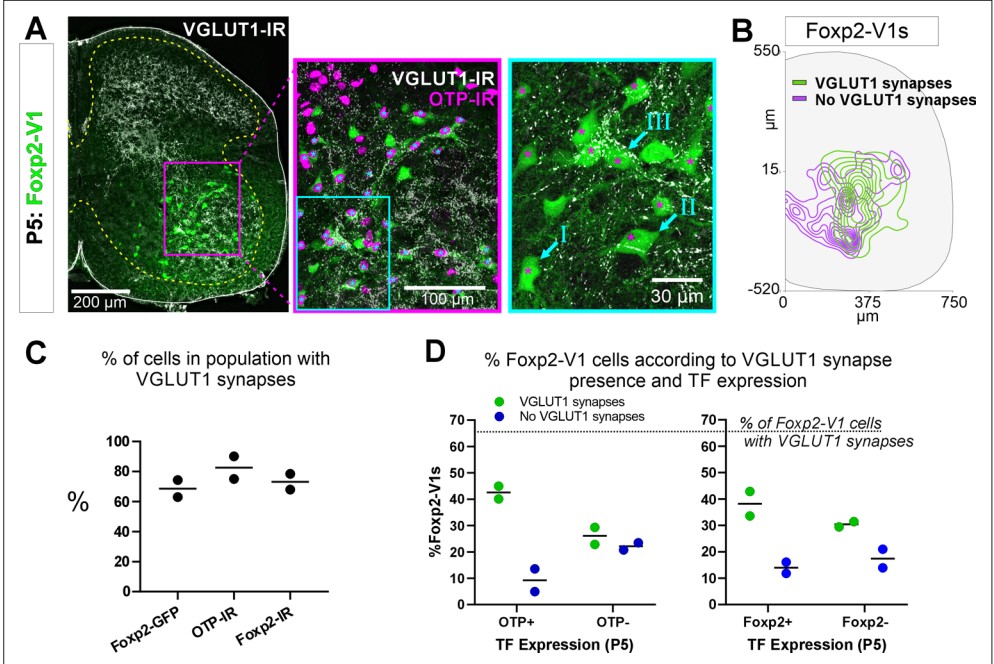

**Figure 7.** Proprioceptive (VGLUT1+) inputs preferentially target lateral Foxp2-V1 interneurons. (**A**) Left, low-magnification confocal image of Foxp2-V1 lineage labeling (EGFP) combined with VGLUT1 antibody staining to identify primary proprioceptive afferent synapses. Center, the boxed area is expanded and shown with the magenta OTP-IR channel overlaying the GFP channel instead of merging with the green to maximize visual discrimination of OTP-IR Foxp2-V1s (denoted by asterisks). The overlay method used here is fully described in Materials and methods, 'Figure composition'. Right, the center image's inlay is shown at high magnification to demonstrate variability of VGLUT1 synapse density on Foxp2-V1 interneurons (I=absent, II = medium or low, III = high). For simplicity and rigor, we classified Foxp2-V1 interneurons as receiving or not receiving VGLUT1 synapses. Asterisks here indicate which of these Foxp2-V1s are OTP-IR based on the lower magnification images. OTP-IR Foxp2-V1s are laterally biased in their positioning and tend to have higher VGLUT1 synapse densities. (**B**) Distribution of Foxp2-V1 interneurons with and without VGLUT1 synapses (green and magenta, respectively). The positioning of Foxp2-V1 interneurons receiving VGLUT1 synapses is laterally biased. (**C**) Percentages of Foxp2-V1s (GFP+ interneurons), OTP-IR Foxp2-V1s, and Foxp2-IR Foxp2-V1s receiving VGLUT1 synapses (both proximal and/or distal). Each dot is an animal estimate from 6 or 7 ventral horns with respectively 591 and 525 Foxp2-V1 interneurons sampled. Lines indicate the averages of both animals. (**D**) Percentages of lineage-labeled Foxp2-V1 interneurons receiving VGLUT1 synapses and with Otp expression (left graph) or Foxp2 (right graph). Each dot represents one mouse, and the lines indicate averages. The numbers of sections and genetically labeled Foxp2-V1 interneurons sampled in each mouse are as in **C**. In each mouse, this includes 237 and 236 Otp-IR cells and 256 and 225 Foxp2-IR cells.

The online version of this article includes the following source data and figure supplement(s) for figure 7:

**Figure supplement 1.** Generation of Otp-Flpo mice and intersection with En1-cre mice to target Otp-V1 cells.

**Figure supplement 1—source data 1.** Raw image of gel corresponding to *Figure 7—figure supplement 1A*.

**Figure supplement 1—source data 2.** Annotated image of gel corresponding to *Figure 7—figure supplement 1A*.

suggests that 68.7% of interneurons in the Foxp2-V1 lineage that were targeted by VGLUT1 synapses had Foxp2 expression at P5, and therefore 73.3% of Foxp2-V1 interneurons with genetic and antibody labeling received VGLUT1 synapses. Of these cells, 60.2% received VGLUT1 inputs proximally. In one of the two animals we also analyzed Otp and Foxp2 co-localization in Foxp2-V1 interneurons receiving VGLUT1+ synapses. We found that 85.6% of Otp+ cells receiving VGLUT1 synapses in the Foxp2-V1 lineage also maintain Foxp2 expression at P5, while 89.8% of genetic and Foxp2 antibody labeled V1s receiving VGLUT1 synapses are also Otp+. In conclusion, laterally positioned Foxp2-V1 interneurons that express Otp and Foxp2 at P5 are preferentially targeted by VGLUT1 synapses.

We attempted to genetically target these cells using an intersection between *Otp* and *En1* with a new *Otp-flpo* mouse (**Figure 7—figure supplement 1A and B** for generation and validation). We found double number of lineage-labeled Otp-V1 cells in the spinal cord of dual-color mice (*En1^{cre/+}*, *Ai9-R26^{lsl-tdT} Otp^{flpo/+}*, R26^{RCE:dual-EGFP}) compared to those V1 cells expressing Otp protein at P5 (**Figure 7—figure supplement 1C and D**). Moreover, only 11.4% of V1 cells expressed only tdT (*En1* and no *Otp*). Both results together suggest widespread transient expression of Otp in V1 cells before P5. Consequently, markers of non-Foxp2-V1 clades, such as calbindin and Pou6f2, were found in significant numbers within lineage-labeled Otp-V1 cells at P5 (**Figure 7—figure supplement 1E and F**). This demonstrates that Otp is expressed by subpopulations of cells in all V1 clades during embryonic development. Unfortunately, currently there is no genetic model to temporally control recombination from the *Otp* locus at P5 and restrict expression to postnatally Otp-expressing Foxp2-V1 cells.

## Foxp2-V1 interneurons form reciprocal inhibitory circuits between antagonistic ankle muscles

Finally, to examine whether Foxp2-V1 interneurons are embedded in reciprocal inhibitory circuits, we combined EGFP lineage labeling of Foxp2-V1 interneurons with anterograde labeling of tibialis anterior (TA) muscle sensory afferents (via cholera toxin subunit B [CTB] muscle injections) and retrograde monosynaptic labeling of interneurons premotor to the antagonistic lateral gastrocnemius (LG) muscle using glycoprotein (G) deleted mCherry rabies virus (RVΔG-mCherry) (**Figure 8A**). The sections were immunolabeled for CTB, VGLUT1, mCherry, and EGFP (**Figure 8B**). To obtain transcomplementation of RVΔG-mCherry with glycoprotein in LG motoneurons, we first injected the LG muscle with an AAV1 expressing B19-G at P4. We then performed RVΔG and CTB injections at P15 to optimize muscle targeting and avoid cross-contamination of nearby muscles. Muscle specificity was confirmed post hoc by dissection of all muscles below the knee. Analyses were done at P22, a time point after developmental critical windows through which Ia (VGLUT1+) synaptic numbers increase and mature on V1-IaINs (*Siembab et al., 2010*).

Unfortunately, motoneuron infection from muscle and transsynaptic retrograde labeling using RVΔG is known to be inefficient after P10 (*Stepien et al., 2010*). Additionally, at older ages transsynaptic transport is slower and temporally spread such that fewer interneurons are recovered at single time points after injection. We chose 7 days post-injection for analyses to avoid as much as possible cell degeneration that frequently occurs at longer survival times after RV infection. We traded the low yield of these experiments for higher specificity when identifying synaptic inputs from TA sensory afferents onto Foxp2-V1 interneurons that are premotor to the LG motor pool. We injected five animals that were analyzed in serial sections from segments L2 to L6. All mice showed consistent TA anterograde labeling that occupied the dorsal third of LVII and LIX in the ventral horn of the L4/L5 segments (**Figure 8B and C**; **Figure 8—figure supplement 1**). This distribution matches the well-known musculotopic trajectories of central Ia afferents axons in the ventral horn (*Ishizuka et al., 1979*). In agreement with the known rostro-caudal distribution of Ia afferent axon collaterals, TA-CTB VGLUT1+ synapses were found in all lumbar segments, but caudal lumbar segments had the largest density in the ventral horn. Additionally, there were dense projections to medial LV and to discrete regions in LIV in all segments and in all animals. Projections to superficial laminae (I to III) were more common in upper lumbar segments. Only three mice showed transsynaptic transport of RVΔG-mCherry from the LG motor pool to interneurons, with large variability from animal to animal. In the best animal, we recovered 51 transsynaptically labeled interneurons with no evidence of degenerative phenotypes (examples in **Figure 8**, **Figure 8—figure supplement 1**). These cells were found at the same locations, and in similar proportions as was reported by other groups using injections in younger animals with more cells were labeled (*Stepien et al., 2010*; *Tripodi et al., 2011*; *Ronzano et al., 2022*). The interneuron sample included cells in the ipsilateral Renshaw area (n=6 or 11.8%), LVII (15, 29.4%), medial LV (7, 13.7%), LI to LIV (20, 39.2%), and the contralateral spinal cord (3, 5.9%: 1 in LX and 2 in LVIII). Pooling cells with transsynaptic labeling from all three animals, we identified 8 out of 15 LVII interneurons as Foxp2-V1. Their dendrites were reconstructed in Neurolucida following mCherry labeling. Five of these cells received more than 1 TA synaptic contact (CTB+ and VGLUT1+) but with large differences in number (5, 9, 12, 17, and 31 synapses). Most synaptic contacts occurred on dendrites, particularly those crossing areas with many CTB-labeled TA afferents. Thus, the direction of dendrites in the section strongly influenced the total number of synapses detected.

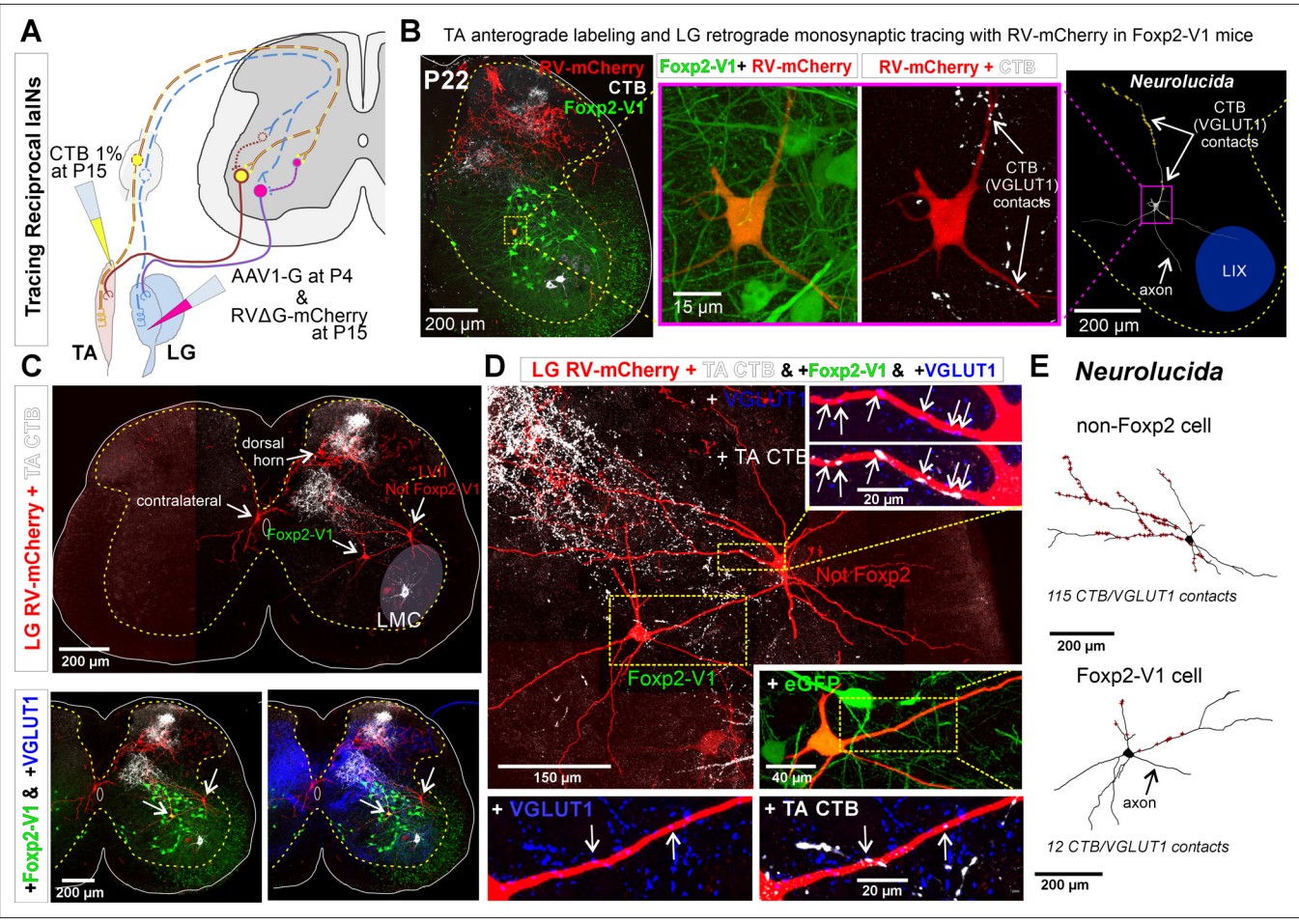

**Figure 8.** Some Foxp2-V1 interneurons are reciprocal Ia inhibitory interneurons (IaINs). (**A**) Experimental design to label spinal neurons that receive inputs from tibialis anterior (TA) muscle primary afferents and connect to lateral gastrocnemius (LG) motoneurons, forming Ia reciprocal inhibitory connections from TA to LG. TA sensory afferents are labeled anterogradely with cholera toxin subunit B (CTB) followed by antibody detection of CTB and the presynaptic marker VGLUT1. Interneurons premotor to LG motoneurons are labeled by monosynaptic retrograde labeling with RVΔG-mCherry. (**B**) Foxp2-V1 IaIN with the most TA/VGLUT1 contacts (31) in our sample (n=5). Left, low-magnification image of Foxp2-V1 interneurons (EGFP, green), RV-mCherry labeling (red) of LG muscle afferents in the dorsal horn and interneurons presynaptic to the LG and of TA afferents anterogradely labeled with CTB (white). The Foxp2-V1 interneuron contains mCherry (yellow cell, inside box). This cell is magnified in two panels to the right, one showing Foxp2-V1 and RV-mCherry and the other RV-mCherry and CTB labeling. Arrows in the zoomed image show examples of CTB synapses (confirmed with VGLUT1) on the dendrites of this neuron. Far right image is the 3D reconstructed cell (Neurolucida) with CTB/VGLUT1 synapses indicated on its dendrites by yellow stars. The axon initial trajectory is indicated (the axon is lost at the section cut surface). The blue area highlights lamina IX. (**C**) Low magnification of section serial to **B**, showing TA-CTB afferents (white) and LG-RV-mCherry-labeled interneurons (red). Transsynaptically labeled interneurons are categorized according to position and Foxp2-V1 lineage labeling: images below show superimposed Foxp2-V1 EGFP (green) and additional VGLUT1 immunolabeling (blue). The location of the LMC is indicated. Contralateral interneurons were found in LX (as the one in this section) and in LVIII (the other two in this animal, not shown). (**D**) High-magnification images of two LVII LG-coupled interneurons (RV-mCherry, red) receiving synapses from TA afferents (dendrites in boxed regions are shown at high magnification demonstrating CTB-TA labeling and VGLUT1 content). The most medial interneuron belongs to the Foxp2-V1 lineage (see inset with +EGFP). Insets shown VGLUT1 synapses with (arrows) and without CTB from the TA muscle. (**E**) Neurolucida neuronal reconstructions showed that the Foxp2-V1 interneuron contained a medium number of TA/VGLUT1 synapses (13 contacts) in our sample of putative IaINs derived from the Foxp2-V1 lineage (n=5), while the non-Foxp2-V1 interneuron contained the largest (115) of any LVII reconstructed interneuron with mCherry, including many proximal synapses. For further examples of labeling in serial sections from this animal, see *Figure 8—figure supplement 1*.

The online version of this article includes the following figure supplement(s) for figure 8:

**Figure supplement 1.** Labeling of motoneurons and interneurons, but only limited labeling of primary afferents with RVΔG-mCherry.

VGLUT1 densities on spinal interneurons are known to depend on dendritic trajectories with respect to VGLUT1 synaptic fields (*Siembab et al., 2016*). The Foxp2-V1 cell shown in *Figure 8B* received the most contacts, which were concentrated on a dendritic segment crossing a field with a high density of TA afferent synapses. Within single sections we found LG-coupled Foxp2-V1 and non-Foxp2 LVII interneurons receiving TA synapses on their dendrites (*Figure 8C and D*). Non-Foxp2 IaINs could be derived from V2b or even non-Foxp2-V1 cells. It is known that several genetic subclasses contribute to the full repertoire of IaINs controlling different leg joints (*Zhang et al., 2014*). These results provide proof-of-principle that some Foxp2-V1 interneurons are in synaptic circuits capable of exerting reciprocal Ia inhibition between antagonistic muscles. Clearly, a technique with higher yield and that also maintains high specificity is necessary. Additionally, analysis of further extensor-flexor pairs in different joints in both directions will need to be performed to reveal a complete picture of IaIN organization.

Other cell types transsynaptically labeled from the LG included V1 Renshaw cells which did not receive any TA synapses since their dendrites are far away from TA projection areas (*Figure 8—figure supplement 1*). Another group were medial LV LG-coupled interneurons (possibly Ia/Ib interneurons not derived from either V2b or V1 classes) which are in a region with high density of TA/VGLUT1+ synapses (*Figure 8—figure supplement 1*). A few reconstructed neurons at this location (n=3) received the highest densities of TA synapses with more than 50 contacts on relatively smaller dendrites. A few other LG-coupled interneurons reside in superficial laminae of the dorsal horn (*Figure 8D*) where they are also contacted by CTB-labeled TA afferent synapses. Muscle afferents ending in superficial laminae are likely non-proprioceptive (i.e. Type III(Aδ) and IV(C) afferents; *Ling et al., 2003*). It has also been shown that interneurons at this location can be transsynaptically labeled in the anterograde direction from sensory afferents (*Zampieri et al., 2014*). However, rabies virus tropism toward primary afferents depends on developmental age, with abundant infection between P0 and P5 and reduced infection by P15. In our experiments, there was very limited mCherry labeling in primary afferents in dorsal roots or dorsal columns. These labeled axons were restricted to upper lumbar segments (*Figure 8—figure supplement 1*, see also *Figure 8B and C*). Thus, ventral mCherry-labeled Foxp2-V1 interneurons in our experiment are most likely transsynaptically labeled in the retrograde direction from a few 'starter' LG motoneurons (*Figure 8—figure supplement 1*).

## Discussion

A comprehensive inventory of spinal premotor interneurons and circuits requires detailed cataloging of their core components, ideally using multimodal information such as genetic subtypes, timing of neurogenesis, settling positions, incorporation within spinal circuits, and their electrophysiological properties and functional roles during motor behaviors. In the present work we focus on V1 inhibitory interneurons, a large heterogenous group of inhibitory interneurons with ipsilateral synaptic projections throughout the ventral horn (*Alvarez et al., 2005*). Previously, the diversity of V1 interneurons in mice was organized into at least four clades deduced from Bayesian statistical analyses of combinatorial expression of 19 TFs and their cell locations in the spinal cord (*Bikoff et al., 2016*; *Gabitto et al., 2016*). Each group divides into further subgroupings organized in a hierarchical fashion. Here, we show that these four V1 clades differ in neurogenesis times and targeting of motoneurons. This result strengthens the conclusion that these V1 clades defined by their genetic make-up represent distinct functional subtypes, although further validation is necessary in more functionally focused studies.

### Sequential neurogenesis of V1 clades reflects their heterogeneous microcircuits

Two V1 clades, Renshaw cells and Pou6f2-V1 cells, have narrow windows of neurogenesis with almost all cells generated before E11, while the Foxp2-V1 and Sp8-V1 clades are generated in a wider temporal window with most born after E11. Within the 'early group' many Renshaw cells are generated before Pou6f2-V1 cells and within the 'late group' Sp8-V1 cells lag behind Foxp2-V1 cells. These data confirm a previous report on the earlier generation of MafB-V1 Renshaw cells compared to Foxp2-V1 IaINs (*Benito-Gonzalez and Alvarez, 2012*). Sequential generation of these two V1 cell types seem to be an intrinsic property of p1 progenitors and can be replicated in vitro using mESC to derive p1 and V1 cells (*Hoang et al., 2018*). The present data extend the view of sequential determination of cell fate by time of neurogenesis to all V1 clades. In addition, the results suggest that clades

with few neurons, limited diversity, and specific locations have narrow windows of neurogenesis, while larger V1 clades, like Foxp2-V1 interneurons, are generated through longer periods of neurogenesis and include subtypes that differ in time of neurogenesis and location.

Overall, our data agree with a previous report that examined spinal cord subtypes based on the intersection of neurogenesis and transcriptomics by analyzing mouse spinal cords from E9.5 to E13.5 (*Delile et al., 2019*). This study divided V1 interneurons into seven groups. Groups V1.5 and V1.2 had early birthdates. The V1.5 group gene expression profile includes *Neurod1, 2 and 4, Neurod1, Prox1, Tcf4, Lhx1 and 5, Pou2f2 and Hes1,* and although it does not clearly match the TF repertoire of the Pou6f2-V1 clade, it occupies a similar birthdate window. In contrast, the V1.2 group shares the gene expression profile of Renshaw cells which includes *Calb1, Mafb,* and *Onecut2*. We consistently found Renshaw cells to be the first born V1 cells, although this study suggested they are generated after the V1.5 group. One explanation is that some of the TFs used to define V1.2 cells, like *Mafb*, are upregulated sometime after neurogenesis (*Benito-Gonzalez and Alvarez, 2012*). The V1.1, V1.4, and V1.3 groups follow in neurogenesis timing, and all three express *Foxp2*; V1.1 and V1.4 have similar birthdates while V1.3 neurogenesis is slightly delayed. We show similar heterogeneity of birthdates in Foxp2-V1 interneurons. Two subgroups expressing the TF Otp at P5 with or without co-expression of Foxp4 and/or Foxp2 have intermediate times of neurogenesis and form a lateral group closely related to LMC motoneurons. They both receive VGLUT1/proprioceptive inputs, and some may form reciprocal inhibitory circuits between antagonistic motor pools. Many Foxp2-V1s located dorsomedially are generated earlier, and Foxp2-V1s located ventromedially are generated last. Many of these cells are not targeted by VGLUT1/proprioceptive inputs which indicates different circuit roles. Finally, groups V1.6 and V1.7 are generated at the end of V1 neurogenesis, and it is tempting to speculate that they include the Sp8-V1 clade and the late generated ventromedial group of Foxp2-V1 interneurons. However, it is presently difficult to match gene expression profiles of V1.6 and V1.7 (*Neurod1, 2*, and *6, Nfia, Nfib, Nfix, Prox1, Tcf4, Hex6, Cbln2, and Slit2*) to our V1 clades. One problem is that the high-throughput sequencing in Delile's paper is based on samples collected at early embryonic times (E9.5 to E13.5), but gene expression in V1 groups changes throughout embryonic and postnatal development. For example, our efforts to generate a genetic model for Otp-V1s demonstrate that the postnatal restriction of this TF to the Foxp2-V1 clade does not occur until after embryogenesis. A similar situation occurs with the Sp8-V1 clade that is best defined by V1 interneurons retaining Sp8 expression at P0 (*Bikoff et al., 2016*), because Sp8 is widely expressed in ventral spinal progenitors during embryogenesis. Conversely, some TFs are not expressed in certain cell types until after neurogenesis or even until late embryonic stages. For example, we obtained preliminary evidence that the medial Foxp2-V1 group upregulates Foxp2 expression after E14 by difference to the lateral group which upregulates Foxp2 expression as they emerge from progenitors (*Benito-Gonzalez and Alvarez, 2012*). Thus, Foxp2 expression might not be captured in medial Foxp2-V1 interneurons when examining gene expression from E9.5 to E13.5. Despite these differences, there is general agreement that Renshaw cells are early born and that several Foxp2-V1 groups are generated later and can be distinguished by differences in neurogenesis.

More generally, our results agree with the pattern of sequentially generated neurons that is consistent throughout the neural tube (summarized by *Sagner, 2024*). Frequently, three neurogenesis waves are described, each defined by specific temporally restricted TFs. The earliest neurons are defined by TFs Onecut1 and Onecut2 which occur in all Renshaw cells, some Pou6f2-V1, a few Foxp2-V1 interneurons and even a subpopulation of Sp8-V1 cells (*Bikoff et al., 2016*). Intermediate neurons express Pou2f2 and Zfhx3 and 4 that are expressed by Pou6f2-V1 and Foxp2-V1 interneurons (Alexandra Trevisan and Jay Bikoff, unpublished data). Finally, late cells express TFs like Nfia, Nfib, Neurod2, and Neurod6 and their location is restricted to medial regions of the neural tube. These correspond to ventromedial subpopulations of Foxp2-V1 cells and most medially located Sp8-V1 interneurons. Thus, expression of temporarily restricted TFs and the locations of cells from the different V1 clades aligns well with the conclusions of our EdU birthdating study: Renshaw cells are earliest generated, Pou6f2-V1 include cells with early and intermediate generation, while Foxp2-V1 and Sp8-V1 cells contain cells that are generated, early, intermediate, or late, at different proportions in each clade.

In addition to differences in neurogenesis timing, we uncovered differences among V1 clades on motoneuron synaptic targeting and this supports the view that these clades constitute unique functional subsets forming distinct ventral horn circuits. Renshaw cells and Foxp2-V1 interneurons are

major sources of V1 synapses on motoneuron cell bodies and proximal dendrites. Pou6f2-V1 and Sp8-V1 interneurons either do not provide a significant direct input to the motoneuron or this occurs quite distally. We previously reported that the density of Sp8-V1 synapses on motoneuron cell bodies and proximal dendrites is between one and two orders of magnitude less dense than Renshaw cell synapses at this location (*Bikoff et al., 2016*). This analysis included a variety of motoneurons inner-vating flexor and extensor muscles at various hindlimb joints, ruling out the possibility that each inter-neuron preferentially targets specific motor pools. Moreover, unpublished data from our group using intracellular fills of dorsal MafB-V1 interneurons belonging to the Pou6f2-V1 clade also show a lack of axon projections toward LIX. Given that V1 axons are largely restricted to the ventral horn (*Alvarez et al., 2005*), it is fair to conclude that Pou6f2-V1 and Sp8-V1 interneurons likely modulate activity in ventral premotor spinal networks and/or perhaps synaptic integration in motoneuron distal dendrites traversing LVII, but do not establish proximal synapses that effectively modulate motoneuron firing and excitability directly. For V1 interneurons that role seems exclusive of Renshaw cells and Foxp2-V1s, although these cells may also target premotor network elements. Within the Foxp2-V1 clade it is not possible at present to define which subgroup provides proximal synapses to motoneurons. However, given the known placement of IaINs (*Jankowska and Lindström, 1972*) and the known proximal location of reciprocal inhibitory synaptic inputs (*Burke et al., 1971*), it is reasonable to expect that many Foxp2-V1 synapses on the cell bodies and proximal dendrites of motoneurons arise from lateral Foxp2-V1 groups (II and III) which occupy spinal cord locations typical of IaINs and preferentially receive VGLUT1/proprioceptive projections.

It is notable that birthdate order was not predictive of presynaptic coupling to motoneurons or phylogenetic relations. Both early born Renshaw cells and late born Foxp2-V1 cells were pref-erentially associated with synapses on motoneuron cell bodies. Moreover, the circuits they form, recurrent and reciprocal inhibition of motoneurons, are both phylogenetically recent. Reciprocal inhibition of muscle flexors and extensors at limb joints first appears in limbed Amphibia, a subclass of tetrapodal vertebrates that lacks recurrent inhibitory circuits (*Czéh, 1977*). At present recur-rent inhibition of motoneurons by Renshaw cells has only been detected in mammals and in the developing hindlimb motor pools of chick embryos (*Wenner and O'Donovan, 1999*; *Alvarez and Fyffe, 2007*). It is thus possible that Renshaw cells and many Foxp2 V1 interneurons are either phylogenetically recent or that these cells acquired novel connectivity in parallel with the evolution of limb motor control. V1 interneurons are found in all vertebrate species ranging from fishes with swimming locomotion to mammals with limbed terrestrial locomotion, but while V1 interneurons show low diversity in zebrafish (*Kimura and Higashijima, 2019*) they display large heterogeneity in mice (*Bikoff et al., 2016*). This could suggest the emergence of novel cell types with the evolution of limbed species. However, the appearance of novel circuits is also supported by observations in the axial MMC, a motor column shared with fishes. These motoneurons are densely innervated by Foxp2-V1s and Renshaw cells and correspondingly, thoracic axial motoneurons in the cat are modulated by Renshaw cell-mediated recurrent inhibition (*Saywell et al., 2013*). This suggests that axial motoneurons in limbed mammals acquired inhibitory controls absent in aquatic vertebrates. Whether this is due to the development of new cell types or the reassignment of existing cells to novel circuits remains to be determined.

The close functional relationship between Renshaw cells and subgroups of Foxp2-V1s involved in reciprocal inhibitory circuits is further emphasized by their intricate connectivity in the mamma-lian spinal cord. Together with flexor and extensor motoneurons, these cells form a canonical motor output module that controls the muscles around individual joints. This canonical microcircuit was first proposed by *Baldissera et al., 1981*, as an end-stage modulator of the spatiotemporal properties of the motor output across all spinal segments in all mammalian species (human and cats at the time, and now also rats and mice). Furthermore, the significance of Renshaw cell and Foxp2-V1 synapses on motoneurons is highlighted by recent studies in ALS mouse models. These studies showed that V1 synapses located on the cell bodies and proximal dendrites of motoneurons are preferentially affected during early stages of the disease, and in addition Foxp2-V1 interneurons have a high susceptibility for cell death (*Wootz et al., 2013*; *Salamatina et al., 2020*; *Montañana-Rosell et al., 2024*). Despite the close relationship between Renshaw cells and Foxp2-V1 interneurons they greatly differ in firing properties and circuit function. Renshaw cells are burst-tonic firing cells, while Foxp2-V1 interneurons are tonic fast-spiking (*Bikoff et al., 2016*) with these properties further diverging during postnatal

maturation. For example, mature Renshaw cell firing properties in postnatal mice (*Bikoff et al., 2016*) greatly differ from early embryos (E12) (*Boeri et al., 2018*; *Boeri et al., 2021*).

In conclusion, previously defined V1 clades exhibit differences in birthdate, heterogeneity, and projections to motoneurons and/or premotor networks. Collectively, these differences may reflect the appearance of novel circuits associated with limb function and overground locomotion. Additionally, early and late born V1 interneurons in the mammalian spinal cord are intricately interconnected and together form critical circuits that modulate motor output.

## Molecular determinants of V1 diversity

In a remarkable in vitro replication of the V1 neurogenesis sequence using mouse embryonic stem cells (mESC), it was found that 24 hr treatment with a high concentration of retinoic acid and a low concentration of smoothened agonist induced mESC derivation of p1 progenitors (*Hoang et al., 2018*). These progenitors sequentially generated V1 interneurons with genetic profiles of Renshaw cells at in vitro day 5 and Foxp2-V1s at day 8. Lengthening the retinoic acid treatment favored differentiation into neurons with Renshaw cell characteristics at the expense of Foxp2-V1s. This suggests that Renshaw cells and Foxp2-V1s derive from a similar pool of p1 progenitors, and that their fates might depend on differences in the morphogenetic signals present at the embryonic times when they are born. For example, in early embryos retinoic acid is highly expressed by the mesoderm adjacent to the developing spinal cord (*Novitch et al., 2003*) and by motoneurons inside the spinal cord (*Sockanathan et al., 2003*), but retinoic acid signals are later attenuated by opposing actions from fibroblast growth factor family members (*Diez del Corral et al., 2003*). A final consideration is that, although Hoang et al. also differentiated V1 cells with gene expression profiles characteristic of Pou6f2 and Sp8 cells, these clades were underrepresented suggesting that p1 progenitors giving rise to these two V1 clades may have distinct signaling requirements during embryogenesis.

## Significance of Foxp2-V1 interneuron diversity

Genetic lineage labeling of Foxp2-V1 interneurons reveals approximately twice as many V1 interneurons compared to the number identified when this clade is defined by postnatal Foxp2 protein expression, because downregulation of Foxp2. Not surprisingly, we uncovered heterogeneity in this clade. We identified four groups of Foxp2-V1s according to TF combinatorial expression and position (*Figure 6G*), and this classification scheme reflected the sequential birthdate order of the groups. Of special interest were groups II and III, which were born at mid-to-late neurogenesis times and expressed Otp at P5. These cells were closely associated with the limb-controlling LMC. Foxp2-V1 interneurons at this location also receive proprioceptive VGLUT1 inputs, and some were found in reciprocal inhibitory circuits between the TA and LG. Proprioceptive TA input densities varied across different dendrites of individual Foxp2-V1 interneurons according to dendrite locations relative to the trajectories of specific primary afferents—a concept already established for the major inputs of Renshaw cells (both proprioceptive and motor) (*Mentis et al., 2006*; *Benito-Gonzalez and Alvarez, 2012*; *Siembab et al., 2016*). Similarly, motoneurons with genetically altered dendritic arbor structure exhibit changes in their proprioceptive inputs (*Vrieseling and Arber, 2006*). In this context, the two groups of Otp-Foxp2-V1 interneurons, defined by Foxp4 expression and dorsoventral location (groups II and III) are likely connected to primary afferents from different muscle groups. In conclusion, the Foxp2-V1 clade represents a diverse group of interneurons. Types II and III may be closely related to proprioceptive pathways, and differ in specificities for particular muscles, joints, and/or flexors and extensors, while types I and IV may be broadly associated with other motor functions.

# Materials and methods

This study was performed in strict accordance with the recommendations in the Guide for the Care and Use of Laboratory Animals of the National Institutes of Health. All of the animals were handled according to approved Institutional Animal Care and Use Committee (IACUC) guidelines at Emory University. The protocol was approved by Emory IACUC (Permit Number: PROTO20170035). All terminal surgeries were performed under deep anesthesia via an overdose of the euthanizing agent Euthasol. All survival surgeries were performed under isoflurane anesthesia with postoperative pain management using buprenorphine. Every effort was made to minimize suffering and postoperative

**Table 1.** Mouse models.

| Mouse | MGI # | RRID # | Brief description | Donating laboratory | Reference |
|---|---|---|---|---|---|
| En1::cre | MGI:3029756 | | En1-cre KI/KO | Goulding (Salk Inst.) | *Sapir et al., 2004* |
| Foxp2::Flpo | MGI:6728072 | | Foxp2-Flpo KI/KO | Bikoff/Jessell (Columbia U.) | *Bikoff et al., 2016* |
| Otp::Flpo | MGI:6728073 | | Otp-Flpo KI/KO | Bikoff/Jessell (Columbia U.) | This manuscript |
| MafB::GFP | MGI:6145008 | RRID:IMSR_RBRC02220 | MafB-GFP KI/KO | Takahashi (Tsukuba U.) | *Moriguchi et al., 2006* |
| Ai9-tdT | MGI:3813511 | RRID:IMSR_JAX:007909 | Rosa26-lsl-tdTom | The Jackson Laboratory | *Madisen et al., 2010* |
| RCE:dual-EGFP | MGI:4420759 | RRID:MMRRC_032036-JAX | Rosa26-lsl-fsf-EGFP | The Jackson Laboratory | *Sousa et al., 2009* |
| RC::FLTG | MGI:5617960 | RRID:IMSR_JAX:026932 | Rosa26-FLTG | The Jackson Laboratory | *Plummer et al., 2015* |
| RCE:FRT | MGI:4420764 | RRID:MMRRC_032038-JAX | RCE-fsf-GFP | Fishell (Harvard) | *Sousa et al., 2009* |

monitoring was conducted according to approved guidelines and recommendations of veterinary staff. ARRIVE E10 guidelines were followed during experimentation and analyses.

## Animal models

To genetically lineage-label all or subclasses of V1 interneurons we used eight transgenic mouse models (*Table 1*). These mice were crossed for intersectional genetic labeling combining a line in which all En1-expressing V1 interneurons express Cre with lines expressing Flpo dependent on Foxp2 or Otp or in which MafB expressing cells express GFP.

### V1 and Foxp2-V1 model

$En1^{Cre/+}$ heterozygotes (*Sapir et al., 2004*) were crossed with *Rosa26-Frt-lox-STOP-lox-tdTomato-WPRE-frt* homozygotes (Ai9 $R26^{lsl-tdT}$, JAX#007909; B6;129S6-Gt(ROSA)26Sort$^{M14(CAG-tdtomato)Hze}$/J) to obtain $En1^{cre/+}$, $Rosa26^{lsl-tdT/+}$ and after backcrossing, $En1^{cre/+}$, $Rosa26^{lsl-tdT/lsl-tdT}$ animals (in both animals all V1 cells are lineage-labeled with tdT). Similarly, $Foxp2^{flpo/+}$ animals (*Bikoff et al., 2016*) were crossed with *RCE:dual-EGFP* homozygotes (RCE:FRT JAX#010675; Gt(ROSA)26Sor$^{tm1(CAG-EGFP)Fsh}$; initially donated by Dr. Gordon Fishell, Harvard University) to produce $Foxp2^{flpo/+}$, $Rosa^{dualEGFPl/+}$ and $Foxp2^{flpo/+}$, $Rosa^{dualEGFP/dualEGFP}$ mice. Crossing $En1^{Cre/+}$, $Rosa26^{lsl-tdT/+}$ or $En1^{cre/+}$, $Rosa26^{lsl-tdT/lsl-tdT}$ mice with $Foxp2^{flpo/+}$ or $Foxp2^{flpo/+}$, $Rosa^{dualEGFP/+}$ or $Foxp2^{flpo/+}$, $Rosa^{dualEGFP/dualEGFP}$ we obtained mice for experiments with the following genotypes: $En1^{cre/+}$, $Foxp2^{flpo/+}$, $Rosa26^{lsl-tdT/+}$ (V1 red), $En1^{cre/+}$, $Foxp2^{flpo/+}$, $Rosa26^{+/dualEGFP}$ (Foxp2-V1 green) or $En1^{cre/+}$, $Foxp2^{flp/+}$, $Rosa26^{lsl-tdT/ dualEGFP}$ (dual color, Foxp2-V1 green and non-Foxp2-V1 red). In some experiments we substituted the reporter lines Ai9 $Rosa26^{lsl-tdT}$ and $Rosa26^{RCE:dualEGFP}$ for the $Rosa26^{RC::FLTG}$ line. This line has in the Rosa26 locus a frt-flanked STOP and loxP-flanked tdTomato::STOP preventing transcription of EGFP (JAX#026932,B6.Cg-Gt(ROSA)26Sor$^{tm1.3(CAG-tdTomato,-EGFP)Pjen}$/J). In this line, Flpo recombination in cells with Foxp2 expression induces expression of tdT, while cells with additional Cre recombination (V1 expressing Foxp2 cells) will express EGFP and remove the tdT reporter.

### V1-Otp model

Similar breeding schemes and reporter lines were used to combine $En1^{cre/+}$ and $Otp^{flpo/+}$ mice to study Otp-V1 interneurons. These mice were generated as described previously (*Bikoff et al., 2016*) and summarized in *Figure 7—figure supplement 1*. Briefly, Flpo, a codon-optimized version of Flp recombinase, was inserted into the ATG in the first exon of the Otp genomic locus, generating a null allele. Positive ES cell clones were screened by Southern blot and microinjected into blastocysts, and the resulting chimeric mice were crossed to C57BL/6J females. The neomycin selectable cassette was removed using Protamine::Cre mice (Jax#03328).

## V1-MafB model

*En1$^{cre/+}$*, *Rosa26$^{lsl-tdT/lsl-tdT}$* animals were crossed with *Mafb$^{GFP/+}$* knock-in mice (**Moriguchi et al., 2006**) to label cells expressing MafB and assess their overlap with V1 interneurons.

The *En1, Foxp2, Otp,* and *Mafb* genes carry the specified transgene knocked in into the gene coding sequence resulting in a null allele. The animals are maintained and bred in heterozygosis, with homozygotes being knockouts for each of these genes. We generated *Foxp2$^{flpo/flpo}$* and *Mafb$^{gfp/gfp}$* homozygotes to test antibody specificities. *Foxp2* knockout mice survive postnatally, but *Mafb* knockouts die at birth due to respiratory failure (**Blanchi et al., 2003**). Thus, *Foxp2* knockout mice were harvested at P5 and *Mafb* knockout mice as late embryos. All animals were bred in our colonies at Emory University, and the resulting litters were genotyped using a combination of standard tail PCR (Transnetyx) and fluorescent phenotyping of neonates (each gene combination results in specific patterns of labeling along the body).

## Timed pregnancies

Female mice were caged with males at the beginning of the dark period (7:00 PM), and the next morning (7:00 AM) vaginal plugs were checked. A positive plug was considered E0.5; however, since the exact time of mating is unknown, this procedure has an approximate error of 12 hr. Moreover, we found embryos within single litters that differ by 6–12 hr in developmental stage.

## Tissue preparation

Mouse pups of different postnatal (P) ages (P0, P5, P15, P30, adult) were anesthetized with an overdose of Euthasol (>100 mg/kg i.p.) and, after transcardial vascular rinsing with phosphate-buffered saline (PBS) and heparin they were perfusion-fixed with 4% paraformaldehyde in 0.1 M phosphate buffer (PB). The spinal cords were then dissected, removed, and post-fixed in 4% paraformaldehyde for either 2 hr, 4 hr, or overnight. Post-fixation times depend on antigens targeted in immunocytochemistry experiments. In general, calcium buffering proteins, choline acetyltransferase, and synaptic vesicle markers require longer post-fixation times while TFs require shorter post-fixation times. After post-fixation, the tissues were cryoprotected in 0.1 M PB with 30% sucrose and prepared for sectioning. Transverse spinal cord sections were obtained in a sliding freezing microtome at 50 µm thickness and collected free-floating. Embryos, P0, and P5 spinal cords were cut in a cryostat at 20 or 30 µm thickness from tissue blocks snap-frozen in OCT.

## Birthdating experiments

EdU (Invitrogen) was injected i.p. at a dose of 50 mg/kg weight in timed-pregnant females. The data reported were obtained from 17 pregnant females successfully injected at gestation days E9.5, 10, 10.5, 11, 11.5, 12, or 12.5 after crossing with appropriate males to generate pups with genetic labels for all V1, Foxp2-V1, or MafB-V1 interneurons. The spinal cords were collected after P5 perfusion-fixation with paraformaldehyde as above. P5 was chosen for analyses to maximize TF antigenicity.

## EdU Click-iT reaction

Fifty micrometer thick transverse spinal cord sections were obtained in a freezing, sliding microtome from lower lumbar segments (4 and 5) and processed free floating with Click-iT Alexa Fluor 488 (C10337, Invitrogen) or Click-iT Alexa Fluor 647 Imaging Kits (C10340, Invitrogen) depending on other genetic fluorophores present in the animal. Sections were washed twice (5 min) with 3% bovine serum albumin (BSA, Fisher) in 0.01 M PBS and then permeabilized with a solution of 0.5% Triton X-100 (Fisher) in 0.01 M PBS at room temperature for 20 min. During this time, the Click-iT reaction cocktail was prepared per the manufacturer's instructions and applied to the sections for 30 min at room temperature protected from light. Sections were washed with 3% BSA in 0.01 M PBS at the conclusion of the incubation. Antibody labeling followed the EdU Click-iT reaction. In one pup EdU labeling did not correspond to the target time point and was discarded (#459.2 in *Figure 1D*). The littermate (#459.1 in *Figure 1D*) displayed correct EdU labeling for the injection time.

## Immunohistochemistry

The characteristics, RRID numbers, and dilutions of all primary antibodies used are summarized in *Table 2*. Genetic labels, tdT and EGFP, were always amplified with antibodies to aid in visualization. After blocking the sections with normal donkey serum (1:10 in PBS+0.3% of Triton X-100; PBS-TX), we incubated the sections in different primary antibody cocktails diluted in PBS-TX. Chicken antibodies were used to detect EGFP, and mouse or rabbit antibodies were used to detect tdT, depending on the hosts of primary antibody combinations. In birthdating experiments we used in serial sections either rabbit anti-MafB (Sigma), goat anti-Foxp2 (Sant Cruz), rabbit anti-Pou6f2 (Sigma), goat anti-Sp8 (Santa Cruz), guinea pig anti-Otp (Jessell Lab; *Bikoff et al., 2016*), rabbit anti-Otp (Jessell Lab; *Bikoff et al., 2016*), rabbit anti-Foxp4 (Jessell Lab; *Bikoff et al., 2016*), or rabbit anti-calbindin (Swant). Depending on color combination for triple or quadruple fluorescent labeling with genetic reporters (EGFP or tdTomato) and the EdU Click-iT reaction (Alexa Fluor 488 or Alexa Fluor 647), these markers were revealed with either FITC-, Cy3-, or Cy5-conjugated species-specific donkey-raised secondary antibodies, or with biotinylated secondary antibodies followed by streptavidin Alexa Fluor 405 (for all secondary antibodies dilution was from 1:100 to 1:200; all secondary reagents were obtained from Jackson ImmunoResearch). After immunoreactions, the sections were mounted on slides and coverslipped with Vectashield (Vector Laboratories). Similar ICC protocols were followed in other experiments.

## Analysis

Confocal images (×10 and ×20) were obtained with an Olympus FV1000 microscope. Image confocal stacks were fed into Neurolucida (MicroBrightField) for counting and plotting cells. We analyzed four ventral horns per animal in lower lumbar segments (L4-L5). Cells were classified according to genetic labeling, TF immunoreactivity, and EdU labeling. EdU-labeled cells were classified as strongly labeled (at least two-thirds of the nucleus uniformly labeled) or weakly labeled (speckles or partial nuclear labeling). From Neurolucida plots we estimated: (1) the percentage of V1 interneurons labeled with EdU (strongly or weakly); (2) the percentage of V1 cells labeled with TF antibodies; (3) the percentage of V1 cells genetically labeled with MafB-GFP; (4) the percentage of V1 interneurons genetically labeled with Foxp2; (5) the percentage of V1 cells with different genetic or ICC markers and incorporating weak or strong EdU at different time points; (6) the cumulative numbers of EdU weakly or strongly labeled cells for all V1 cells and for each marker across all embryonic times.

## Cell location/density analyses

From Neurolucida plots we constructed cell density profiles for each V1 interneuron type and birthdate, assigning Cartesian coordinates to the nucleus location with respect to the dorsal edge of the central canal which was defined as position (0,0). Coordinates were exported as .csv files and plotted using custom MATLAB scripts to display the position of each individual cell (*Bikoff et al., 2016*). Distribution contours were constructed in MATLAB using the kde2d function (MATLAB File Exchange), which estimates a bivariate kernel density over a set of grid points. We plotted density contours containing from 5% (inner contours) to 95% (most outer contour) of the cell population in 10% increments. Density contours were superimposed onto hemicord schematic diagrams in which the distance from central canal to lateral, dorsal, or ventral boundaries was adjusted depending on age and segment from measurements obtained in Neurolucida.

The MATLAB script deposited in GitHub, copy archived at *Worthy, 2024*.

## Analyses of lineage-labeled Foxp2-V1, Otp-V1, and MafB(GFP)-V1 cells across ages and markers

The spinal cords of animals carrying different combinations of genetic labels were prepared as above for amplification with immunolabeling of their fluorescent reporters and combination with different markers (for the characteristics of the different samples in terms of number of animals and sections analyzed see the text in Results).

**Table 2.** Antibodies.

| Antigen | Immunogen | Host/type | Manufacturer | Catalog # | RRID # | Dilution |
|---|---|---|---|---|---|---|
| OTP | Recombinant protein [DPGGHPGDLAPNSDPVEGATC] | Guinea Pig, polyclonal | Jessell Lab/ HHMI CU | CU1497 | RRID:AB_2665423 | 1:3000 |
| OTP | Recombinant protein (aa1–130 from human OTP) | Rabbit, polyclonal | Thermo Fisher | PA5-89060 | RRID:AB_2805328 | 1:2000 |
| Pou6f2 | Recombinant protein (Pou6f2) | Rabbit, polyclonal | Sigma | hpa008699 | RRID:AB_1079664 | 1:1000 |
| Pou6f2 | Synthetic peptide amino acids 35–184 of human Pou6f2 | Guinea Pig, polyclonal | Jessell Lab/ HHMI CU | | RRID:AB_2665423 | 1:500 |
| Pou6f2 | Synthetic peptide amino acids 35–184 of human Pou6f2 | Rat, monoclonal | Jessell Lab/ HHMI CU | | RRID:AB_2665427 | 1:1000 |
| Sp8 | Synthetic peptide (C-terminus [C-18] from human Sp8) | Goat, polyclonal | Santa Cruz | sc-104661 | RRID:AB_2194626 | 1:2000 |
| Foxp2 | Synthetic peptide (N-terminus [N-16] from human FOXP2) | Goat, polyclonal | Santa Cruz | sc-21069 | RRID:AB_2107124 | 1:2000 ICC and WB |
| Foxp4 | Recombinant protein [DPGGHPGDLAPNSDPVEGATC] | Rabbit, polyclonal | Jessell Lab/ HHMI CU | CU1464 | RRID:AB_2665415 | 1:25000 |
| MafA | Recombinant protein (aa300–359 [C-terminus] from mouse v-mafA) | Rabbit, polyclonal | Novus | NB400-137 | RRID:AB_10002142 | 1:500 WB |
| MafB | Recombinant protein (Transcription factor MafB) | Rabbit, polyclonal | Sigma | hpa005653 | RRID:AB_1079293 | 1:1000 ICC and WB |
| MafB | Recombinant protein (aa100–150 from mouse MafB) | Rabbit, polyclonal | Novus | NB600-266 | RRID:AB_2137664 | 1:250 ICC 1:500 WB |
| c-Maf | Recombinant protein (aa150–200 from mouse c-Maf) | Rabbit, polyclonal | Novus | NB600-267 | RRID:AB_2137514 | 1:500 WB |
| NeuroD2 | Synthetic peptide Mouse NeuroD2 aa1–100 conjugated to KLH | Rabbit, polyclonal | AbCam | Ab104430 | RRID:AB_10975628 | 1:2000 |
| Prox1 | Synthetic peptide from the C-terminus of mouse Prox1 | Rabbit, polyclonal | Millipore Sigma | AB5475 | RRID:AB_177485 | 1:500 |
| RFP | Recombinant protein (RFP from *Discosoma* sp.) | Mouse, monoclonal | Rockland | 200-301-379 | RRID:AB_2611063 | 1:1000 |
| dsRed | Recombinant protein (RFP variant from *Discosoma* sp.) | Rabbit, polyclonal | Clontech | 632496 | RRID:AB_10013483 | 1:1000 |
| GFP | Recombinant protein (GFP from *Aequorea victoria*) | Chicken, polyclonal | Aves | GFP-1020 | RRID:AB_10000240 | 1:1000 |
| ChAT | Human placental enzyme | Goat, polyclonal | EMD Millipore | AB144P | RRID:AB_2079751 | 1:100 |
| NeuN | Synthetic peptide of human RBFOX3/NeuN protein (aa20–100) | Rabbit, polyclonal | Novus | NBP1-77686 | RRID:AB_11009597 | 1:1000 |
| NeuN | Isolated brain cell nuclei | Mouse clone A60 | Millipore | MAB377 | RRID:AB_2298772 | 1:100 |
| Calbindin | Recombinant protein (calbindin D-28k from rat) | Rabbit, polyclonal | Swant | CB-38a | RRID:AB_10000340 | 1:1000 |
| VGLUT1 | Synthetic peptide (aa456–560 from rat VGLUT1) | Guinea Pig/ polyclonal | Synaptic Systems | 135-304 | RRID:AB_887878 | 1:1000 |
| VGAT | Synthetic peptide (N-terminus from rat VGAT) | Mouse/ monoclonal | Synaptic Systems | 131-011 | RRID:AB_887872 | 1:100 |
| VGAT | Recombinant protein (N-terminus from rat VGAT) | Guinea Pig/ polyclonal | Synaptic Systems | 131-004 | RRID:AB_887873 | 1:200 |
| CTB | Cholera toxin B subunit | Goat/ polyclonal | List Labs | #703 | RRID:AB_10013220 | 1:200 |
| acetyl-Histone H3 | Linear peptide from human Histone H3 acetylated at the N-terminus | Rabbit/ polyclonal | EMD Millipore | 06-599 | RRID:AB_2115283 | 1:1000 |

## Analysis of Foxp2-V1 interneurons in different segments of the adult spinal cord

We used *En1^{cre/+}*, *Foxp2^{flpo/+}*, *Rosa26^{+/dualEGFP}* mice and amplified the EGFP signal as above. The position of motoneurons was revealed using a goat anti-ChAT (Millipore) antibody. Laminae cytoarchitectonics were assessed with a rabbit recombinant anti-NeuN antibody (Novus) (not shown in figures). ChAT and NeuN immunoreactivities were detected with Cy3- and Cy5-conjugated donkey anti-goat IgG antibodies, respectively (Jackson ImmunoResearch). Confocal microscopy images were obtained (as above), plotted in Neurolucida, and cell density contours generated. We analyzed the number of Foxp2-V1 cells and cholinergic motoneurons per section and calculated their ratios (a motoneuron was defined as any cholinergic immunoreactive cell in LIX). Density contours were used to compare distributions in different segments.

## Analyses of TF expression in Foxp2-V1 interneurons at P5

To examine Otp and Foxp4 immunoreactivity in lineage-labeled Foxp2 vs non-Foxp2 V1 cells, we used dual-color *En1^{cre/+}*, *Foxp2^{flp/+}*, *Rosa26^{lsl-tdT/dualEGFP}* mice. The sections were immunolabeled with guinea pig anti-Otp (Jessell Lab, CU) and rabbit anti-Foxp4 (Jessell Lab, VU). Otp immunoreactive sites were revealed with biotinylated donkey anti-guinea pig followed by streptavidin Alexa Fluor 405 and Foxp4 immunoreactivity was detected with donkey anti-rabbit IgG secondary antibody conjugated to Cy5. These antibodies were combined with ICC amplification of the EGFP signal (chicken anti-EGFP) and tdTomato (mouse anti-RFP) respectively with FITC- and Cy3-conjugated species-specific IgG antibodies raised in donkey. To analyze the populations of lineage-labeled Foxp2-V1 cells expressing Otp, Foxp4, and Foxp2 at P5, we used single-color *En1^{cre/+}*, *Foxp2^{flp/+}*, *Rosa26^{+/dualEGFP}* mice and amplified EGFP fluorescence with antibodies as above, combined with guinea pig anti-Otp (Jessell Lab, CU)/ rabbit anti-Foxp4 (Jessell Lab, CU), rabbit anti-Otp (Thermo Fisher)/goat anti-Foxp2 (Santa Cruz), and rabbit anti-Foxp4 (Jessell Lab, CU)/goat anti-Foxp2 (Santa Cruz) antibodies. Triple immunostains were revealed with species-specific secondary antibodies as above using the Cy3 and Cy5 channels for TFs. A few sections were immunolabeled with rabbit antibodies against NeuroD2 (Abcam) or Prox1 (Millipore/Sigma). In this case immunoreactivities were revealed with Cy3-conjugated anti-rabbit IgG antibodies. All sections were imaged using confocal microscopy and confocal stacks analyzed in Neurolucida as above. All analyses were done in Lumbar 4 and 5 segments.

## Analyses of V1 clade markers in MafB-V1 and Otp-V1 interneurons at P5

For these analyses we used *En1^{cre/+}*, *Otp^{flpo/+}*, *Rosa26^{+/dualEGFP}* or *En1^{cre/+}*, *Rosa26^{lsl-tdT/+}*, *Mafb^{GFP/+}* mice. To analyze Otp-V1 interneurons, EGFP fluorescence was amplified with chicken anti-EGFP antibodies as above, and one of the following additional primary antibodies was added in serial sections: rabbit anti-calbindin (Swant), guinea pig anti-Otp (Jessell Lab, CU), guinea pig anti-Pou6f2 (Jessell Lab, CU), goat anti-Foxp2 (Santa Cruz), or goat anti-Sp8 (Santa Cruz). All markers were detected in the Cy3 channel. When possible, they were combined with goat anti-ChAT antibodies (EMD Millipore) or mouse anti-NeuN antibodies (Millipore) labeled in the Cy5 channel. ChAT and NeuN immunoreactivities are not shown in Results but were used to identify laminae and spinal cord segments according to motor column organization and cytoarchitectonics. In MafB-V1 mice we amplified both EGFP and tdT as above and combined with the following antibodies: rabbit anti-calbindin (Swant), guinea pig anti-Pou6f2 (Jessell Lab, CU), goat anti-Foxp2 (Santa Cruz), and goat anti-Sp8 (Santa Cruz). All marker antibodies were revealed using Cy5-conjugated donkey species-specific anti-IgG secondary antibodies as above. Analyses were done as above: confocal images were imported into Neurolucida for cell plotting and the results expressed as number or proportion of neurons and their position analyzed using cell distribution density profiles. All analyses were done in Lumbar 4 and 5 segments.

## Analyses of Foxp2-V1 and Foxp2-non-V1 cells at P5

For these analyses we generated two *En1^{cre/+}*, *Foxp2^{flpo/+}*, *Rosa26^{+/FLTG}* mice. EGFP (Foxp2-V1 cells) and tdTomato (Foxp2-non-V1 cells) were amplified with antibodies as above. The sections were counterstained with mouse anti-NeuN antibodies (Millipore) for segment and laminar localization. Sections from Lumbar 4 and 5 segments were imaged with confocal microscopy and analyzed in Neurolucida. Cell distribution plots, cell numbers, and cell density curves were obtained as above.

**Table 3.** Cell lines.

| Cell line | Brief description/validation | Donating laboratory | Reference |
|---|---|---|---|
| B7GG | B7GG cells express T7 RNA polymerase, rabies virus G, and histone-tagged GFP. Their origin is BHK-21 cells. They express GFP when expressing RV-G. Cell producing virus have additionally mCherry in the cytoplasm. | Dr. Edward Callaway (Salk Institute, La Jolla, CA, USA) | *Osakada and Callaway, 2013* |
| HEK 293T-TVA800 cells | HEK 293T-TVA800 cells are derived from HEK 293T cells and express TVA. These cells are used to titer pseudotyped G-deleted rabies viruses. | Dr. Edward Callaway (Salk Institute, La Jolla, CA, USA) | *Osakada and Callaway, 2013* |

## Analyses of VGLUT1 inputs on Foxp2-V1 interneurons

Analyses were done at P5 in *En1*$^{cre/+}$, *Foxp2*$^{flp/+}$, *Rosa26*$^{dualEGFP/+}$ mice. The P5 age was selected to preserve TF immunoreactivity. Moreover, at this age VGLUT1 synapses in the ventral horn are specifically contributed by proprioceptive afferents, most likely Ia afferents (*Alvarez et al., 2004*). For these analyses spinal cord sections were obtained in a cryostat (20 μm thickness). EGFP was amplified with chicken anti-GFP (Aves) and combined with goat anti-Foxp2 (Santa Cruz), rabbit anti-Otp (Thermo Fisher), and guinea-pig anti-VGLUT1 (Synaptic Systems). EGFP, Foxp2, and Otp immunoreactivities were revealed respectively with FITC-, Cy3-, and Cy5-conjugated species-specific secondary antibodies. VGLUT1 was revealed with biotinylated anti-guinea pig IgG antibodies followed by streptavidin Alexa Fluor 405. The sections were imaged at ×10 and ×60 with confocal microscopy. VGLUT1 contacts were analyzed only in high-magnification images in which we tiled the whole ventral horn to sample every Foxp2-V1 cell present. Using Neurolucida, we plotted all Foxp2-V1 cells in each section and classified them according to the presence of VGLUT1 inputs and whether they were at high or low density in a qualitative assessment. Later we pooled all data by the presence or absence of VGLUT1 contacts. From these plots we estimated: (1) the percentage of lineage-labeled Foxp2-V1 interneurons with VGLUT1 contacts; (2) the percentage of these cells with Otp, Foxp2, or both TFs co-localized (for sample attributes with respect to number of animals, sections, and cells analyzed, see Results).

## Identification of Foxp2-V1 interneurons interposed in reciprocal connections between the TA muscle and the LG motor pool

For these analyses we combined retrograde monosynaptic tracing of RVΔG-mCherry from the LG muscles with anterograde tracing of muscle sensory afferent synapses using CTB from the TA. RVΔG-mCherry and CTB intramuscular injections were done at P15 to avoid critical windows of synaptic reorganization. To facilitate tracing at this age we applied RVΔG-mCherry at high titer (>10$^9$ TU/ml) and CTB was injected at high concentration (1%). RVΔG-mCherry was produced in the lab using SADB19ΔG-mCherry rabies virus (RVΔG-mCherry samples and cells lines donated by Dr. Edward Callaway, Salk Institute, La Jolla, CA, USA) (see *Table 3* for description of cell lines).

### Production of RVΔG-mCherry

Sterile cell culture technique without antibiotics was used throughout all procedures. B7GG cells were placed in cell culture dishes containing DMEM in 10% fetal bovine serum (FBS) (culture medium) and incubated in 5%$CO_2$ in humid air at 37°C for 2 hr. Once they adhered to the substrate, the medium was removed, the cultures washed with 10 ml warm PBS, and fresh medium applied. Four plates were grown to 90% confluency, and then 4 ml of virus stock was added to each culture and incubated at 37°C/5%$CO_2$ for 4 hr. After washing three times in warm PBS (to remove as much virus as possible) the cultures were incubated in fresh medium at 35°C/3%$CO_2$ and monitored daily for fluorescence. Four plates of fresh B7GG cells were grown to 90% confluency, washed with 10 ml PBS, and detached with 6 ml of 0.25% trypsin for 5 min at room temperature with gentle rocking. A warm culture medium with 10% FBS was added to quench trypsin activity and the cells were dissociated by gentle trituration (approximately 12 passes). The cell suspensions were centrifuged at 800×*g* for 3 min to pellet cells. After removing the trypsin/culture medium they were re-suspended in warm culture medium. Twelve culture dishes containing 18 ml of culture medium were simultaneously inoculated with 2 ml each of the B7GG cell suspension and incubated at 37°C/5%$CO_2$. The culture medium was changed

after 80–90% confluency and 4 ml of viral supernatant added to each plate and incubated for 4 hr at 37°C/5%CO$_2$, then culture medium was removed, the cells washed three times in PBS and 20 ml fresh medium added to each plate. The cultures were incubated at 35°C/3%CO$_2$ and monitored daily for expression of cytoplasmic mCherry and nuclear GFP. The cell culture medium containing RVΔG-mCherry was collected after 4 days, filtered through 0.45 µm membranes, and placed on ice. RVΔG-mCherry was concentrated from 180 ml (10 plates) of cell supernatant via ultracentrifugation, 2 hr at 20,000×$g$ at 4°C. The supernatant was aspirated, and the six viral pellets were re-suspended in 200 µl each of cold Hanks Balanced Salt Solution (HBSS). All supernatants were combined and layered over 1.8 ml of 20% sucrose in HBSS in a 3 ml tube. The virus was pelleted through the sucrose cushion by ultracentrifugation, at 20,000×$g$ for 2 hr at 4°C. The supernatant and sucrose cushion were gently poured off and any remining fluid aspirated. The pellet was re-suspended in 105 µl of HBSS by gentle agitation and 5 µl aliquots frozen at –80°C for later use.

## Virus titer

Serial dilutions of RVmCH (from one 5 µl aliquot of virus) from the preparation were inoculated onto 1 × 10$^5$ HEK-293T-TVA800 cells in a 48-well plate. As soon as mCherry was detectable (usually 2 days), the positive cells in the well (dilution) containing 10–100 cells were counted and a titer obtained. For these experiments we used the rabies virus at a titer of 1.92×E10 transfection units (TU)/ml.

## Production of AAV1-G

Adeno-associated viruses were produced by the Emory Virus Core from AAV2 plasmids expressing the B19 RV glycoprotein under a CMV promoter. This plasmid was donated by Dr. Silvia Arber (Biozentrum, Basel) (*Stepien et al., 2010*). Virus titer was expressed in genomic copies (GC) and was 2.5×E12 vg/ml by qPCR for the lot used here.

## Viral and tracer intramuscular injections

AAV1-G was injected at P4 in the LG (1–2 µl, undiluted), RVΔG-mCherry was injected in the same muscle at P15 (2–3 µl, undiluted) and CTB was injected in the TA at the same time (0.5 µl, 1% diluted is sterile saline). The animals were allowed to survive 7 days after the last injection (P22) at which time they were perfusion-fixed, and 50-µm-thick sections were prepared as above.

## Immunocytochemistry

Sections were incubated overnight in a cocktail of primary antibodies that included chicken anti-GFP (Aves), rabbit anti-DsRed (Clontech), guinea pig anti VGLUT1 (Synaptic Systems) and goat anti-CTB (List Labs). Primary antibodies were detected with species-specific antibodies coupled to FITC (for EGFP), Cy3 (for tdT), Cy5 (for CTB), or biotin (for VGLUT1). Biotinylated antibodies were exposed with streptavidin Alexa Fluor 405.

## Analyses

All sections with mCherry-labeled cells were imaged at low (×10), medium (×20), and high (×60) magnification using confocal microscopy. Reciprocal IaINs were defined as neurons retrogradely labeled from the LG by RVΔG-mCherry and receiving inputs from TA CTB/VGLUT1-labeled boutons. Because this technique was relatively low yield. All analyses were qualitative. The numbers of animals and yields are reported in Results.

## Antibody specificities

The most frequently used antibodies in this study were tested in knockout mice: guinea pig anti-VGLUT1 and rabbit anti-calbindin (*Siembab et al., 2010*), rabbit anti-Pou6f2, goat anti-Sp8, and guinea pig anti-Otp (*Bikoff et al., 2016*), and goat anti-Foxp2 and two rabbit anti-MafB antibodies (this study, *Figure 1—figure supplements 1 and 2*). In addition, Foxp2 and MafB antibodies used here were further characterized using western blots (see below). Alternative antibodies against Pou6f2 or Otp were first confirmed in dual immunolabeling with validated antibodies. NeuroD2 and Prox1 antibodies were tested and used in recent literature (*Osseward et al., 2021*). Goat anti-ChAT antibodies exhibited the well-known patterns of cholinergic immunoreactive neurons in the spinal cord

and coincided with genetically labeled neurons in ChAT-IRES-Cre-tdTomato mice. GFP, DsRed, and RFP antibodies did not result in any immunolabeling in sections not expressing any of these fluorescent proteins. CTB antibodies resulted in no staining in naïve sections. Rabbit and mouse anti-NeuN antibodies gave identical results. The immunostaining of the mouse anti-NeuN monoclonal in the spinal cord has been amply characterized (*Alvarez et al., 2004*; *Alvarez et al., 2005*).

## Western blots

Foxp2 and MafB antibodies were further characterized in western blots from spinal cord samples collected from wildtypes, heterozygotes (one null allele), and homozygous knockouts (both null alleles) (*Figure 1—figure supplements 1 and 2*). Lumbar spinal cords were dissected in oxygenated artificial cerebrospinal fluid and immediately homogenized using Cytoplasmic Extraction Reagent Kit from the NE-PER nuclear extraction kit (Thermo Fisher) with a protease inhibitor cocktail added (5 mg/ml, Complete Mini, Roche). The manufacturer's instructions were followed to isolate nuclei and yield aliquots of nuclear proteins. A Bio-Rad DC protein assay was used to determine total nuclear protein content. Protein standards (Bio-Rad) were prepared in NER buffer from the NE-PER kit. Sample absorbance was read on a plate reader at 750 nm. Samples were stored at –80°C until use. For immunoblotting, the samples were prepared in standard SDS-PAGE sample buffer (5×) and 30 µl of nuclear protein from each spinal cord was added to each lane of a Bio-Rad precast 10% polyacrylamide gel. Bio-Rad Kaleidoscope molecular weight markers were added to one lane. Electrophoresis was carried out in Tris buffer (Bio-Rad) at 180 V until the dye front reached the bottom of the gel. The proteins were then transferred overnight onto PVDF membranes (Bio-Rad) using standard SDS-PAGE transfer buffer (Bio-Rad) and a constant 0.15 A current with gentle stirring at 4°C. For MafB antibodies, three immunostaining procedures were carried out successively on the same membrane. The membrane was washed three times with Tris Buffered Saline and Triton X-100 (TBST) for 15 min and blocked with non-fat dry milk (Carnation, Nestle) for 1 hr at room temperature on a rocker. The first immunostaining employed a primary antibody against MafB (Novus) at a dilution 1:500 and worked best with 2.5% NFDM in TBST and incubated overnight at 4°C. The primary antibody was detected with donkey anti-rabbit IgG secondary antibodies conjugated to horseradish peroxidase (HRP, GE Health Sciences) and immunostained bands were revealed using enhanced chemiluminescence (ECL). The blot was then stripped for 20 min (ReBlot Plus), washed three times for 15 min in TBS, blocked in 5% NFDM for 1 hr and re-probed with a primary antibody against MafA (Novus) at 1:500 in TBST and the immunoreactive bands detected via ECL as above. Finally, the blot was stripped a final time using ReBlot Plus (Millipore) for 20 min, blocked in 5% NFDM, and re-probed with an antibody against MafB (Sigma). Secondary antibodies and ECL were identical to the previous two primary antibodies. In a different sequence we substituted the MafB (Sigma) antibody for c-Maf (Novus). Similar procedures were used to detect Foxp2 immunoreactivity (Santa Cruz) except that in this case the blot was probed only once, and we used anti-goat secondary antibodies coupled to HRP. Images of western blots are not cropped and show the full gel.

## Statistical analyses

All statistical analyses were performed in Prism (GraphPad v.9). In all cases the samples passed the normality test. When comparing multiple groups, we used one- or two-way ANOVAs depending on the sample structure. To include consideration of repetitive measures in single animals (e.g. different motoneurons from single animals) we used nested one-way ANOVAs. Post hoc pairwise comparisons were always done using Bonferroni-corrected t-tests. When comparisons involved only two groups we used standard t-tests. All statistical details are provided in *Supplementary file 1*.

## Figure composition

All images for presentation were obtained with an Olympus FV1000 confocal microscope and processed with Image-Pro Plus (Media Cybernetics) for optimization of image brightness and contrast. Frequently we used a high Gaussian filter to increase sharpness. Figures were composed using CorelDraw (v.X6 and CDR2023). Graphs were generated in Prism (GraphPad, v.9). Color choices were always selected to optimal visualization by people with different degrees of color detection capacity. For *Figure 7* we decided to superimpose the OTP signal in magenta and avoid merging the magenta color with the green EGFP which have resulted in a difficult to detect whitish green. Instead, we added

the magenta on top of the green neurons. These image manipulations were performed in Image Pro-Plus by first subtracting the OTP+ labeled nucleus from the EGFP image and then adding the magenta OTP signal. Because magenta is restricted to the nucleus the EGFP (green) cells are easily detectable and the OTP+ cells within this group are best visualized.

## Acknowledgements

We want to thank Zoë Haley-Johnson and Indera Cogdell for their help in maintaining these colonies. This research project was supported in part by the Viral Vector Core of the Emory Center for Neuro-degenerative Disease Core Facilities. Funding: This work was supported by the NIH-NINDS grant R01 NS047357 to FJA.

## Additional information

### Funding

| Funder | Grant reference number | Author |
| --- | --- | --- |
| National Institutes of Health | R01 NS047357 | Francisco J Alvarez |

The funders had no role in study design, data collection and interpretation, or the decision to submit the work for publication.

### Author contributions

Andrew E Worthy, Alicia R Lane, Data curation, Formal analysis, Investigation, Writing – review and editing; Joanna T Anderson, Laura J Gomez-Perez, Anthony A Wang, Formal analysis, Investigation; Ronald W Griffith, Resources, Methodology; Andre F Rivard, Investigation, Methodology; Jay B Bikoff, Resources, Writing – review and editing; Francisco J Alvarez, Conceptualization, Resources, Data curation, Formal analysis, Supervision, Funding acquisition, Validation, Investigation, Visualization, Methodology, Writing - original draft, Project administration, Writing – review and editing

### Author ORCIDs

Andrew E Worthy ⓘ https://orcid.org/0000-0002-9992-2675
Alicia R Lane ⓘ https://orcid.org/0000-0002-6404-7559
Francisco J Alvarez ⓘ https://orcid.org/0000-0001-7011-3244

### Ethics

This study was performed in strict accordance with the recommendations in the Guide for the Care and Use of Laboratory Animals of the National Institutes of Health. All of the animals were handled according to approved IACUC guidelines at Emory University. The protocol was approved by Emory IACUC (Permit Number: PROTO20170035). All terminal surgeries were performed under deep anesthesia via an overdose of the euthanizing agent Euthasol. All survival surgeries were performed under isoflurane anesthesia with postoperative pain management using buprenorphine. Every effort was made to minimize suffering and postoperative monitoring was conducted according to approved guides and recommendations of veterinary staff.

Reviewer #1 (Public review): https://doi.org/10.7554/eLife.95172.3.sa1
Reviewer #2 (Public review): https://doi.org/10.7554/eLife.95172.3.sa2
Author response https://doi.org/10.7554/eLife.95172.3.sa3

## Additional files

### Supplementary files

• Supplementary file 1. Statistical figures for the data presented. Contains all the statistical tables for graphs shown in figures. 1a is related to *Figure 4D*; 1b is related to *Figure 5D*; 1c is related to *Figure 5E*, top graph; 1d is related to *Figure 5E*, bottom graph; 1e is related to *Figure 6C*, left

graph Otp/Foxp2; 1f is related to *Figure 6C*, center graph Foxp4/Foxp2; 1g is related to *Figure 6C*, right graph Otp/Foxp4.

• MDAR checklist

### Data availability

The quantitative data generated in this study is available at Emory Dataverse. Details of statistics and sample sizes and organization are presented in tables in *Supplementary file 1*. Images and Neurolucida datasets: This manuscript is based on analyses of around 1,250 confocal images, sometimes consisting of tiled images with many more sub-images within. These images were collected over the expanse of 12 years (2011-2023) by AW, JTA, ARL, LGT, AAW and FJA. Some of these images contain information that continues to be analyzed in the Alvarez lab in ongoing projects. Readers can request any image sets or corresponding Neurolucida cell analyses by emailing directly the corresponding author. Gels: Raw images of all gels presented in this paper are submitted as source data files. The modified code form the MATLAB script used in the analysis of cell densities is deposited in GitHub, copy archived at *Worthy, 2024*.

The following dataset was generated:

| Author(s) | Year | Dataset title | Dataset URL | Database and Identifier |
|---|---|---|---|---|
| Alvarez F | 2024 | Data for Worthy et al., Spinal V1 inhibitory interneuron clades differ in birthdate, projections to motoneurons and heterogeneity Elife 2024 | https://doi.org/10.15139/S3/CXG9FB | Emory Dataverse, 10.15139/S3/CXG9FB |

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
