## [Editor Report · eLife Assessment]

This study provides a **valuable** description of subtypes of V1 neurons, including birthdates and connections to motor neurons. V1 neurons are one of the main groups of inhibitory neurons in the spinal cord. The methods of data collection and analysis are **convincing**. This work will interest developmental biologists and neuroscientists working on spinal circuits.

---

## [Referee Report · Reviewer #1 (Public review)]

To understand spinal locomotor circuits, we need to reveal how various types of spinal interneurons work in them. So far, the general roles of the cardinal groups of spinal interneurons (dI6, V0, V1, V2a, V2b, and V3) in locomotion have been studied but not fully understood. Each group is believed to contain some subgroups with more detailed functional differences. However, each character and function of these subgroups has yet to be elucidated.

In this study, Worthy et al. investigated V1 neurons, one of the main groups of inhibitory neurons in the spinal cord. Previous reports proposed four major clades in V1 neurons defined by the expression of transcription factors (MafA/MafB, Foxp2, sp8, and pou6f2). The authors investigated the birth time for V1 neurons in each of the four clades and showed the postnatal location in the spinal cord with different birthdates. Next, the authors investigated the Foxp2-V1 population in detail using genetically labeled Foxp2-V1 mice. They found some FoxP2-V1 located near LMC motor neurons that innervate limbs. They showed that most of the synapses of V1 neurons on the cell bodies of LMC motor neurons were from Foxp2-V1 and Renshaw cells, and the proportion of Foxp2-V1 synapses in V1 synapses on motor neurons was relatively high in LMC compared to other motor columns. They also proposed that Foxp2-V1 can be further classified according to the expression of transcription factors Otp and Foxp4. The results of this paper are well supported by the data obtained using widely used methods.

This study will be helpful for future analyses of the development and function of V1 neurons. In particular, the discovery of strong synaptic connections between Foxp2-V1 and LMC motor neurons will be beneficial in analyzing the role of V1 neurons in motor circuits that generate movement of the limbs.

---

## [Referee Report · Reviewer #2 (Public review)]

Summary:

This work brings important information regarding the composition of interneurons in the mammalian spinal cord, with a developmental perspective. Indeed, for the past decades, tools inspired from developmental biology have opened up promising avenues for challenging the functional heterogeneity in the spinal cord. They rely on the fact that neurons sharing similar mature properties also share a largely similar history of expression of specific transcription factor (TF) genes during embryogenic and postnatal development. For instance, neurons originating from p1 progenitors and expressing the TF Engrailed-1, form the V1 neuronal class. While such "cardinal" neuronal classes defined by one single RF indeed share numerous features - e.g., for the case of V1 neurons, a ventral positioning, an inhibitory nature and ipsilatetal projections - there is accumulating evidence for a finer-grained diversity and specialization in each class which is still largely obscure. The present work studies the heterogeneity of V1 interneurons and describes multiple classes based on their birthdate, final positioning, and expression of additional TF. It brings in particular a solid characterization of the Foxp2-expressing V1 interneurons for which authors also delve into the connectivity, and hence, possible functional implication. The work will be of interest to developmental biologists and those interested in the organization of the locomotor spinal network.

Strengths:

This study has deeply analyzed the diversity of V1 neurons by intersecting multiple criteria: TF expression, birthdate, location in the spinal cord, diversity along the rostro-caudal axis, and for some subsets, connectivity. This illustrates and exemplifies the absolute need to not consider cardinal classes, defined by one single TF, as homogeneous. Rather, it highlights the limits of single-TF classification and exemplifies the existence of further diversity within the cardinal class.

Experiments are generally well performed with a satisfactory number of animals and adequate statistical tests.

Authors have also paid strong attention to potential differences in cell-type classification when considering neurons currently expressing of a given TF (e.g., using antibodies), from those defined as having once expressed that TF (e.g., defined by a lineage-tracing strategy). This ambiguity is a frequent source of discrepancy of findings across studies.

Furthermore, there is a risk in developmental studies to overlook the fact that the spinal cord is functionally specialized rostro-caudally, and to generalize features that may only be applicable to a specific segment and hence to a specific motor pool. While motoneurons share the same dorso-ventral origin and appear homogenous on a ChAT staining, specific clusters are dedicated to specific muscle groups, e.g., axial, hypaxial or limb muscles. Here, the authors make the important distinction between different lumbar levels and detail the location and connectivity of their neurons of interest with respect to specific clusters of MN.

Finally, the authors are fully transparent on inter-animal variability in their representation and quantification. This is crucial to avoid the overgeneralization of findings but to rather provide a nuanced understanding of the complexities of spinal circuits.

Weaknesses:

The different V1 populations have been investigated in detail regarding their development and positioning, but their functional ambition is not directly investigated through gain or loss of function experiments in the present study. While the putative inputs onto motoneurons are interesting and suggestive of differences between V1 pools, they are only a little predictive of function.

---

## [Author Response]

The following is the authors’ response to the original reviews.

**Reviewer 1:**
Comment 1. In Figure 1, the MafB antibody (Sigma) was used to identify Renshaw cells at P5. However, according to the supplementary Figure 3D, the specificity of the MafB antibody (Sigma) is relatively low. The image of MafB-GFP, V1-INs, and MafB-IR at P5 should be added to the supplementary figure. The specificity of MaFB-IR-Sigma in V1 neurons at P5 should be shown. This image also might support the description of the genetically labeled MafB-V1 distribution at P5 (page 8, lines 28-32).

We followed the reviewer’s suggestion and moved analyses of the MafB-GFP mouse to a supplemental figure (Fig S3). The characterization of MafB immunoreactivities is now in supplemental Figure S2 and the related text in results was also moved to supplemental to reduce technicalities in the main text. We added confocal images of MafB-GFP V1 interneurons at P5 showing immunoreactivities for both MafB antibodies, as suggested by the reviewer (Fig S2A,B). We agree with the reviewer that this strengthens our comparisons on the sensitivity and specificity of the two MafB antibodies used in this study.

As explained in the preliminary response we cannot show lack of immunoreactivity for MafB antibodies in MafB GFP/GFP knockout mice at P5 because MafB global KOs die at birth. This is why we used tissues from late embryos to check MafB immunoreactivities (Figure S2C and S2D). We made this point clearer in the text and supplemental figure legends.

Comment 2. The proportion of genetically labeled FoxP2-V1 in all V1 is more than 60%, although immunolabeled FoxP2-V1 is approximately 30% at P5. Genetically labeled Otp-V1 included other nonFoxP2 V1 clades (Fig. 8L-M). I wonder whether genetically labeled FoxP2-V1 might include the other three clades. The authors should show whether genetically labeled FoxP2-V1 expresses other clade markers, such as pou6f2, sp8, and calbindin, at P5.

We included the requested data in Figure 3E-G. Lineage-labeled Foxp2-V1 neurons in our genetic intersection do not include cells from other V1-clades.

**Reviewer 2:**
Comment 1. The current version of the paper is VERY hard to read. It is often extremely difficult to "see the forest for the trees" and the reader is often drowned in methodological details that provide only minor additions to the scientific message. Non-specialists in developmental biology, but still interested in the spinal cord organization, especially students, might find this article challenging to digest and there is a high risk that they will be inclined to abandon reading it. The diversity of developmental stages studied (with possible mistakes between text and figures) adds a substantial complexity in the reading. It is also not clear at all why authors choose to focus on the Foxp2 V1 from page 9. Naively, the Pou6f2 might have been equally interesting. Finally, numerous discrepancies in the referencing of figures must also be fixed. I strongly recommend an in-depth streamlining and proofreading, and possibly moving some material to supplement (e.g. page 8, and elsewhere).

The whole text was re-written and streamlined with most methodological discussion (including the section referred to by the reviewer) transferred to supplemental data. Nevertheless, enough details on samples, stats and methods were retained to maintain the rigor of the manuscript.

The reasons justifying a focus on Foxp2-V1 interneurons were fully explained in our preliminary response. Briefly, we are trying to elucidate V1 heterogeneity, and prior data showed that this is the most heterogeneous V1 clade (Bikoff et al., 2016), so it makes sense it was studied further. We agree that the Pou6f2 clade is equally interesting and is in fact the subject of several ongoing studies.

Comment 2. … although the different V1 populations have been investigated in detail regarding their development and positioning, their functional ambition is not directly investigated through gain or loss of function experiments. For the Foxp2-V1, the developmental and anatomical mapping is complemented by a connectivity mapping (Fig 6s, 8), but the latter is fairly superficial compared to the former. Synapses (Fig 6) are counted on a relatively small number of motoneurons per animal, that may, or may not, be representative of the population. Likewise, putative synaptic inputs are only counted on neuronal somata. Motoneurons that lack of axo-somatic contacts may still be contacted distally. Hence, while this data is still suggestive of differences between V1 pools, it is only little predictive of function.

We fully answered the question on functional studies in the preliminary response. Briefly, we are currently conducting these studies using various mouse models that include chronic synaptic silencing using tetanus toxin, acute partial silencing using DREADDs, and acute cell deletion using diphtheria toxin. Each intervention reveals different features of Foxp2-V1 interneuron functions, and each model requires independent validation. Moreover, these studies are being carried out at three developmental stages: embryos, early postnatal period of locomotor maturation and mature animals. Obviously, this is all beyond the goals and scope of the present study. The present study is however the basis for better informed interpretations of results obtained in functional studies.

Regarding the question on synapse counts, we explained in the preliminary results fully why we believe our experimental designs for synapse counting at the confocal level are among the most thorough that can be found in the literature. We counted a very large number of motoneurons per animal when adding all motor column and segments analyzed in each animal. Statistical power was also enough to detect fundamental variation in synaptic density among motor columns.

We focus our analyses on motoneuron cells bodies because analysis of full dendritic arbors on all motor columns present throughout all lumbosacral segments is not feasible. Please see Rotterman et al., 2014 (J. of Neuroscience; doi: 10.1523/JNEUROSCI.4768-13.2014) for evaluation of what this entails for a single motoneuron. We agree with the reviewer that analyses of V1 synapses over full dendrite arbors in specific motoneurons will be very relevant in further studies. These should be carried out now that we know which motor columns are of high interest. Nevertheless, inhibitory synapses exert the most efficient modulation of neuronal firing when they are on cell bodies, and our analyses clearly suggest a difference in in cell body inhibitory synapses targeting between different V1 interneuron types that we find very relevant.

Comment 3. I suggest taking with caution the rabies labelling (Figure 8). It is known that this type of Rabies vectors, when delivered from the periphery, might also label sensory afferents and their postsynaptic targets in the cord through anterograde transport and transneuronal spread (e.g., Pimpinella et al., 2022). Yet I am not sure authors have made all controls to exclude that labelled neurons, presumed here to be premotoneurons, could rather be anterogradely labelled from sensory afferents.

Over the years, we performed many extensive controls and validation of rabies virus transsynaptic tracing methods. These were presented at two SfN meetings (Gomez-Perez et al., 2015 and 2016; Program Nos. 242.08 and 366.06). Our validation of this technique was fully explained in our preliminary response. We also pointed out that the methods used by Pimpinella et al. have a very different design and therefore their results are not comparable to ours. In this study we injected the virus at P15 into leg muscles, and not directly into the spinal cord. In our hands, and as cited in Pimpinella et al., the rabies virus loses tropism for primary afferents with age when injected in muscle. The lack of primary afferent labeling in key lumbosacral segments (L4 and L5) is now illustrated in a new supplemental figure (Figure S6). This figure also shows some starter motoneurons. As explained in the text and in our previous response, these are few in number because of the reduced infection rate when using this method in mature animals (after P10).

Comment 4. The ambition to differentiate neuronal birthdate at a half-day resolution (e.g., E10 vs E10.5) is interesting but must be considered with caution. As the author explains in their methods, animals are caged at 7pm, and the plug is checked the next morning at 7 am. There is hence a potential error of 12h.

We agree with the reviewer, and we previously explicitly discussed these temporal resolution caveats. We have now further expanded on this in new text (see middle paragraph in page 5). Nevertheless, the method did reveal the temporal sequence of neurogenesis of V1 clades with close to 12-hour resolution.

As explained in text and preliminary response this is because we analyzed a sufficient number of animals from enough litters and utilized very stringent criteria to count EdU positives.

Moreover, our results fit very well with current literature. The data agree with previous conclusions from Andreas Sagner group (Institut für Biochemie, Friedrich-Alexander-Universität Erlangen-Nürnberg), on spinal interneurons (including V1s) birthdates based on a different methodology (Delile J et al. Development. 2019 146(12):dev173807. doi: 10.1242/dev.173807. PMID: 30846445; PMCID: PMC6602353). In the discussion we compared in detail both the data and methods between Delile article and our results. We also cite Sagner 2024 review as requested later in the reviewer’s detailed comments. Our results also confirmed our previous report on the birthdates of V1-derived Renshaw cells and Ia inhibitory interneurons (Benito-Gonzalez A, Alvarez FJ J Neurosci. 2012 32(4):1156-70. doi: 10.1523/JNEUROSCI.3630-12.2012. PMID: 22279202; PMCID: PMC3276112). Finally, we recently received a communication notifying us that our neurogenesis sequence of V1s has been replicated in a different vertebrate species by Lora Sweeney’s group (Institute of Science and Technology Austria; direct email from this lab) and we shared our data with them for comparison. This manuscript is currently close to submission. Therefore, we are confident that despite the limitations of EdU birthdating we discussed, the conclusions we offered are strong and are being validated by other groups using different methods and species. We also want to acknowledge the positive comments of reviewer 3 regarding our birthdating study, indicating it is one the most rigorous he or she has ever seen.

**Reviewer 3:**
Comment 1. My only criticism is that some of the main messages of the paper are buried in technical details. Better separation of the main conclusions of the paper, which should be kept in the main figures and text, and technical details/experimental nuances, which are essential but should be moved to the supplement, is critical. This will also correct the other issue with the text at present, which is that it is too long.

Similar to our response to comment 1 from Reviewer 2 we followed the reviewers’ recommendations and greatly summarized, simplified and removed technical details from the main text, trying not to decrease rigor.

**Reviewer #1 (Recommendations For The Authors):**

In Figure 1, the definition of the area to analyze MafB ventral and MafB dorsal is unclear. It should be described.

This has been clarified in both text and supplemental figure S3.

“We focused the analyses on the brighter dorsal and ventral MafB-V1 populations defined by boxes of 100 µm dorsoventral width at the level of the central canal (dorsal) or the ventral edge of the gray matter (ventral) (Supplemental Figure S3B).”

Problems with figure citation.

We apologize for the mistakes. All have been corrected.

**Reviewer #2 (Recommendations For The Authors):**
As indicated in the public review, I'd recommend to substantially revise the writing, for clarity. As such, the paper is extremely hard to read. I would also recommend justifying the focus on Foxp2 neurons.Also, the scope of the present paper is not clearly stated in the introduction (page 4).

Done. We also modified the introduction such that the exact goals are more clearly stated.

I would also recommend toning down the interpretation that V1 clades constitute "unique functional subsets" (discussion and elsewhere). Functional investigation is not performed, and connectomic data is partial and only very suggestive.

We include the following sentence at the end of the 1st paragraph in the discussion:

“This result strengthens the conclusion that these V1 clades defined by their genetic make-up might represent distinct functional subtypes, although further validation is necessary in more functionally focused studies.”

Different post-natal stages are used for different sections of the manuscript. This is often confusing, please justify each stage. From the beginning even, why is the initial birthdating (Figure 1) done here at p5, while the previous characterization of clades was done at p0? I am not sure to understand the justification that this was chosen "to preserve expression of V1 defining TFs". Isn't the sooner the better?

The birthdating study was carried out at P5. P5 is a good time point because there is little variation in TF expression compared to P0, as demonstrated in the results. Furthermore, later tissue harvesting allows higher replicability since it is difficult to consistently harvest tissue the day a litter is born (P0). Also technically, it is easier to handle P5 tissue compared to P0. The analysis of VGUT1 synapses was also done at P5 rather than later ages. This has two advantages: TFs immunoreactivities are preserved at this age, and also corticospinal projections have not yet reached the lumbar cord reducing interpretation caveats on the origins of VGUT1 synapses in the ventral horn (although VGLUT1 synapses are still maturing at this age, see below).

Other parts of the study focus on different ages selected to be most adequate for each purpose. To best study synaptic connectivity, it is best to study mature spinal cords after synaptic plasticity of the first week. For the tracing study we thoroughly explain in the text the reasons for the experimental design (see also below in detailed comments). For counting Foxp2-V1 interneurons and comparing them to motor columns we analyze mature animals. For testing our lineage labeling we use animals of all ages to confirm the consistency of the genetic targeting strategy throughout postnatal development and into adulthood.

Figure 5: wouldn't it be worth quantifying and illustrating cellular densities, in addition to the average number of Foxp2 neurons, across lumbar segments (panel D & E)? Indeed, the size of - and hence total number of cells within - each lumbar segment might not be the same, with a significant "enlargement" from L2 to L4 (this is actually visible on the transverse sections). Hence, if the total number of cells is in the higher in these enlarged segments, but the total number of Foxp2-V1 is not, it may mean that this class is proportionally less abundant.

We believe the critical parameter is the ratio of Foxp2-V1s to motoneurons. This informs how Foxp2-V1 interneurons vary according to the size of the motor columns and the number of motoneurons overall.

The question asked by the reviewer would best be answered by estimating the proportion of Foxp2-V1 neurons to all NeuN labeled interneurons. This is because interneuron density in the spinal cord varies in different segments. We are not sure what this additional analysis will contribute to the paper.

Why, in the Rabies tracing scheme (Fig 8), the Rabies injection is performed at p15? As the authors explain in the text, rabies uptake at the neuromuscular junction is weak after p10. It is not clear to me why such experiments weren't done all at early postnatal stages, with a "classical" co-injection of TVA and Rabies.

First, we do not need TVA in this experiment because we are using B19-G coated virus and injecting it into muscles, not into the spinal cord directly.

Second, enhanced tracing occurs when the AAV is injected a few days before rabies virus. This is because AAV transgene expression is delayed with respect to rabies virus infection and replication. We have performed full time courses and presented these data in one abstract to SfN: Gomez-Perez et al., 2015 Program Nos. 242. We believe full description of these technical details is beyond the scope of this manuscript that has already been considered too technical.

Third, the justification of P15 timing of injections for anterograde primary afferent labeling and retrograde monosynaptic labeling of interneurons is fully explained in the text.

“To obtain transcomplementation of RVDG-mCherry with glycoprotein in LG motoneurons, we first injected the LG muscle with an AAV1 expressing B19-G at P4. We then performed RVDG and CTB injections at P15 to optimize muscle targeting and avoid cross-contamination of nearby muscles. Muscle specificity was confirmed post-hoc by dissection of all muscles below the knee. Analyses were done at P22, a timepoint after developmental critical windows through which Ia (VGLUT1+) synaptic numbers increase and mature on V1-IaINs (Siembab et al., 2010)”

Furthermore, CTB starts to decrease in intensity 7 days after injection because intracellular degradation and rabies virus labeling disappears because cell death. Both limit the time of postinjection for analyses.

Likewise, I am surprised not to see a single motoneuron in the rabies tracing (Fig 8), neither on histology nor on graphs (Fig 8). How can authors be certain that there was indeed rabies uptake from the muscle at this age, and that all labelled cells, presumed to be preMN, are not actually sensory neurons? It is known that Rabies vectors, when delivered from the periphery, might also label sensory afferents and their post-synaptic targets through anterograde transport and transneuronal spread (e.g., Pimpinella et al., 2022). This potential bias must be considered.

This is fully explained in our previous response to the second reviewer’s general comments. We have also added a confocal image showing starter motoneurons as requested (Figure S6A).

Please carefully inspect the references to figures and figure panels, which I suspect are not always correct.

Thank you. We carefully revised the manuscript to correct these deficiencies and we apologize for them.

**Reviewer #3 (Recommendations For The Authors):**
Figure 1: Data here is absolutely beautiful and provides one of the most thorough studies, in terms of timepoints, number of animals analyzed, and precision of analysis, of edU-based birth timing that has been published for neuron subtypes in the spinal cord so far. My only suggestion is to color code the early and late born populations (in for example, different shades of green for early; and blue for late, to better emphasize the differences between them). It is very difficult to differentiate between the purple, red and black colors in G-I, which this would also fix. The antibody staining for Pou6f2 (F) is also difficult to see; gain could be increased on these images or insets added for clarity.

The choice of colors is adapted for optimal visualization by people with different degrees of color blindness. Shades of individual colors are always more difficult to discriminate. This is personally verified by the senior corresponding author of this paper who has some color discrimination deficits. Moreover, each line has a different symbol for the same purpose of easing differentiation.

Figure 2: This is also a picture-perfect figure showing further diversity by birth time even within a clade. One small aesthetic comment is that the arrows are quite unclear and block the data. Perhaps the contours themselves could be subdivided by region and color coded by birth time-such that for example the dorsal contours that emerge in the MafB clade at E11 are highlighted in their own color. Some quantification of the shift in distribution as well as the relative number of neurons within each spatially localized group would also be useful. For MafB, for example, it looks as though the ventral cells (likely Renshaw) are generated at all times in the contour plots; in the dot plots however, it looks like the most ventral cells are present at e10.5. This is likely because the contours are measuring fractional representations, not absolute number. An independent measure of absolute number of ventral and dorsal, by for example, subdividing the spinal cord into dorsoventral bins, would be very useful to address this ambiguity.

We believe density plots already convey the message of the shift in positions with birthdate. We are not sure how we can quantify this more accurately than showing the differences in cellular density plots. We used dorsoventral and mediolateral binning in our first paper decades ago (Avarez et al., 2005). This has now been replaced by more rigorous density profiles that describe better cell distributions. Unfortunately, to obtain the most accurate density profiles we need to pool all cells from all animals precluding statistical comparisons. This is because for some groups there have very few cells per animal (for example early born Sp8 or Foxp2 cells).

Figure 3 and Figure 4: These, and all figures that compare the lineage trace and antibody staining, should be moved to the supplement in my opinion-as they are not for generalist readers but rather specialists that are interested in these exact tools. In addition, the majority of the text that relates to these figures should be transferred to the supplement as well. Figure 5: Another great figure that sets the stage for the analysis of FoxP2V1-to-MN synaptic connectivity, and provides basic information about the rostrocaudal distribution of this clade, by analyzing settling position by level. I have only minor comments. The grid in B obscures the view of the cells and should be removed. The motor neuron cell bodies in C would be better visible if they were red.

We moved some of the images to supplemental (see new supplemental Fig S4). However, we also added new data to the figure as requested by reviewers (Fig 3E-G). We preserved our analyses of Foxp2 and non-Foxp2 V1s across ages and spinal segments because we think this information is critical to the paper. Finally, we want to prevent misleading readers into believing that Foxp2 is a marker that is unique to V1s. Therefore, we also preserved Figures 3H to 3J showing the non-V1 Foxp2 population in the ventral horn.

Figure 6: Very careful and quantitative analysis of V1 synaptic input to motor neurons is presented here. For the reader, a summary figure (similar to B but with V1s too) that schematizes V1 FoxP2 versus Renshaw cell connectivity with LMC, MMC, and PGC motor neurons are one level would be useful.

Thanks for the suggestion. A summary figure has now been included (Figure 5G).

Figure 7: The goal of this figure is to highlight intra-clade diversity at the level of transcription factor expression (or maintenance of expression), birth timing and cell body position culminating in the clear and concise diagram presented in G. In panels A-F however, it takes extra effort to link the data shown to these I-IV subtypes. The figure should be restructured to better highlight these links. One option might be to separate the figure into four parts (one for each type): with the individual spatial, birth timing and TF data for each population extracted and presented in each individual part.

We agree with the reviewer that this is a very busy figure. We tried to re-structure the figure following the suggestions of the reviewer and also several alternative options. All resulted in designs that were more difficult to follow than the original figure. We apologize for its complexity, but we believe this is the best organization to describe all the data in the simplest form.

Figure 8: in A-D, the main point of the figure - that V1FoxP2Otp preferentially receive proprioceptive synapses is buried in a bunch of technical details. To make it easier for the reader, please:(1) add a summary as in B of the %FoxP2-V1 Otp+ cells (82%) with Vglut1 synapses to make the point stronger that the majority of these cells have synapses.

We added this graph by extending the previous graph to include lineage labeled Foxp2-V1s with OTP or Foxp2 immunoreactivity. It is now Figure 7B.

(2) Additionally, add a representative example that shows large numbers of proximal synapses on an FoxP2-V1 Otp+.

The image we presented before as Figure 8A was already immunostained for OTP, so we just added the OTP channel to the images. Now all this information is in panels that are subparts of Figure 7A.

(3) Move the comparison between FoxP2-V1 and FoxP2AB+V1s to the supplement.

We preserved the quantitative data on Foxp2-V1 lineage cells with Foxp2-immunoreactivity but made this a standalone figure, so it is not as busy.

(4) Move J-M description of antibody versus lineage trace of Otp to supplement as ending with this confuses the main message of the paper (see comment above).

All results for the Otp-V1 mouse model have now been placed in a supplemental figure (Figure 5S).

Discussion: A more nuanced and detailed discussion of how the temporal pattern of subtype generation presented here aligns with the established temporal transcription factor code (nicely summarized in Sagner 2024) would be helpful to place their work in the broader context of the field.

This aspect of the discussion was expanded on pages 20 and 21. We replaced the earlier cited review (Sagner and Briscoe, 2019, Development) with the updated Sagner 2024 review and further discussed the data in the context of the field and neurogenesis waves throughout the neural tube, not only the spinal cord. We previously carefully compared our data with the spinal cord data from Sagner’s group (Delile et, 2019, Development). We have now further expanded this comparison in the discussion.